# Spongy all-in-liquid materials by in-situ formation of emulsions at oil-water interfaces

Parisa Bazazi [1✉], Howard A. Stone [2] & S. Hossein Hejazi [1✉]

Printing a structured network of functionalized droplets in a liquid medium enables engineering collectives of living cells for functional purposes and promises enormous applications in processes ranging from energy storage to tissue engineering. Current approaches are limited to drop-by-drop printing or face limitations in reproducing the sophisticated internal features of a structured material and its interactions with the surrounding media. Here, we report a simple approach for creating stable liquid filaments of silica nanoparticle dispersions and use them as inks to print all-in-liquid materials that consist of a network of droplets. Silica nanoparticles stabilize liquid filaments at Weber numbers two orders of magnitude smaller than previously reported in liquid-liquid systems by rapidly producing a concentrated emulsion zone at the oil-water interface. We experimentally demonstrate the printed aqueous phase is emulsified in-situ; consequently, a 3D structure is achieved with flexible walls consisting of layered emulsions. The tube-like printed features have a spongy texture resembling miniaturized versions of "tube sponges" found in the oceans. A scaling analysis based on the interplay between hydrodynamics and emulsification kinetics reveals that filaments are formed when emulsions are generated and remain at the interface during the printing period. Stabilized filaments are utilized for printing liquid-based fluidic channels.

[1] Department of Chemical and Petroleum Engineering, University of Calgary, Calgary, AB T2N 1N, Canada. [2] Department of Mechanical and Aerospace Engineering Princeton University, Princeton, NJ 08544, USA. ✉email: parisa.bazazi@ucalgary.ca; shhejazi@ucalgary.ca

Liquid-in-liquid printed materials[1–6] have many potential applications in energy storage[7,8], microreactors[9], and for creating biomimetic materials[10–14]. These types of materials can be generated by application of an electrical field[15], using molding[4,16], or direct ink writing (DIW) printing techniques[1,6]. Liquid-fluid interfaces can be arrested in desired non-equilibrium shapes by the adsorption of colloidal particles. For instance, bicontinuous structures, called bijels, are formed from the spinodal decomposition of a binary liquid mixture containing amphiphilic particles[17,18]. Spherical bubbles covered with particles[19] and liquid droplets coated with the assembly of nanoparticles and end-functionalized polymers[15] can be redesigned into various anisotropic shapes by the jamming of the particles at the interface, caused by the reduction of the surface area. Nanoparticle jamming at liquid-liquid interfaces can prevent jet breakup[6], where stabilized liquid filaments are used to create all-in-liquid composite materials of oil-water[1,3,4,20–22] and water-water[2,23]. The particle jamming technique is applicable for a wide range of liquid viscosities and many particle-polymer combinations, and enables the exchange of materials between the printed texture and the surrounding fluid. However, the materials generated with the particle jamming approach lack multiscale porosity created, for example, by emulsion-based inks used in 3D printed solid structures[24–26].

3D printing techniques, in combination with the polymerization-induced phase separation, enable the formation of microstructures within the printed solid structures in air[27]. In these methods, after printing a multicomponent mixture in a desired configuration, the polymerization starts and initiates the phase segregation. Internal textures are created within the printed frame during polymerization as they get solidified[27,28]. Structured liquids with submicrometer domains can also be achieved by the solvent transfer induced phase separation (STRIPS) approach[29,30]. A homogeneous oil-water-alcohol-surfactant-particle mixture is rapidly injected into a continuous water phase. The phase separation starts upon the injection, where alcohol is extracted into the continuous phase from the mixture. The presence of particles results in the arrest of the mixture in a transition zone of bicontinuous structures during the phase separation[30]. Recently, internal structures inside the all-in-liquid printed materials were achieved with photocurable polymers[31]. Oil-water interfaces can be stabilized by the interactions of surfactant assemblies, i.e., micelles, with fatty acids, where micelle morphology is changed from spherical to lamella and a gel phase is formed[31,32]. In the presence of photocurable polymers inside the internal phase, submicrometer regions are formed after photopolymerization[31]. The STRIPS approach is limited to the use of a ternary mixture and the photopolymerization method eliminates the fluidity of the printed material. Besides, the aforementioned techniques do not form an interconnected network of droplets similar to those with emulsions. Thus, a technique for creating spontaneous and droplet-based 3D printed all-in-liquid materials, i.e., self-assembled compartmental emulsification, is required.

Emulsions allow encapsulating hydrophilic and hydrophobic cargoes together into the same printed texture[33,34]. Also, emulsions enable interactions between the printed frame and the surrounding media, where the space within the printed texture, i.e., porosity, is controlled by the droplet size distribution[24–26]. Although emulsions are ideal colloidal dispersions for printing in air, their utilization in liquid printing systems is limited to sticky (gel-like) emulsions[35]. The stability of the emulsion is the main challenge that hinders their applications as inks in liquid media. Either the injected emulsion phase is immediately dispersed in the surrounding liquid, or the droplets swell and coalescence. Consequently, conventional emulsions are not appropriate candidates for printing in liquid media.

We describe a new approach to print liquids in a liquid medium by the formation of an emulsion zone at the oil-water interface. Silica nanoparticles are incorporated into spontaneous emulsification systems of sorbitan monooleate (Span 80) micellar solutions. Upon contacting an oleic micellar solution with the aqueous silica dispersion, tiny droplets nucleate inside the micellar solution close to the oil–water interface, forming an emulsion phase. The rapid expansion of the interfacial area due to the low oil-water interfacial tension creates interconnected structures of oil and emulsion phases. The spontaneous formation of an emulsion phase at the oil-water interface and its further penetration inside the bulk stabilizes liquid filaments with walls made of submicrometer emulsion droplets that can be used as inks for liquid-in-liquid printing.

## Results

**Morphological states of liquid columns**. The aqueous phase (deionized (DI) water or 4.0 wt.% silica dispersion) is injected with an average speed $8.0 \times 10^{-4}$–$1.1 \times 10^{-1}$ m/s, calculated as the injection flow rate divided by the cross-sectional area of the needle, into a reservoir of mineral oil containing a 20.0 wt.% Span 80 micellar solution. The oil viscosity is 135 mPa.s, while the addition of Span increases the oil viscosity to 257 mPa s (material properties in Supplementary Note 1, Figs. S1, S2). For the case of DI water, single drops detach from the injection needle tip and sediment through the micellar solution, as is familiar from the Rayleigh-Plateau instability (Fig. 1a and Supplementary Note 2, Figs. S3–S6). The injection of a 4.0 wt.% silica dispersion generates three flow regimes (Fig. 1b). These flow morphologies were characterized by a fast Fourier transform analysis (Supplementary Note 3, Figs S7–S12) and named bead-on-a-string (BOAS), column, and connected, and color coded respectively in blue, green, and red throughout the manuscript (Fig. 1 and Supplementary Movies 1–4).

The BOAS regime appears for the lowest injection speeds, where the detached drops merge with the preceding drops, and is identified with the dominant wavelength of 1.9 mm < $\lambda_{\mathrm{mean}}$ < 2.6 mm. Please note that the column and connected flow regimes do not have a dominant wavelength (Supplementary Note 3). Thus, we consider the cut-off wavelength, where the power spectrum starts decreasing, as the characteristic wavelength in the column and connected flow regimes. In the column state, which occurs for intermediate injection speeds, the injected liquid forms a stable thread-like shape without breaking into droplets, where the cut-off wavelength is 0.7 mm < $\lambda_{\mathrm{cut-off}}$ < 1.0 mm. The connected regime occurs at high injection speeds, where a thread connects the neighboring droplets, i.e., a combination of column and BOAS shapes, with wavelengths of 2.5 mm < $\lambda_{\mathrm{cut-off}}$ < 3.2 mm (Fig. 1b, c). Here, our particular interest is on identifying conditions that lead to the column regime since stable columns have potential applications for direct ink writing, as we demonstrate below.

Additional experiments are conducted over a range of silica concentrations 0.0, 1.0, 2.0, and 4.0 wt.% and Span concentrations 1.0, 5.0, 10.0, 20.0 and 40.0 wt.% (Figs. S3–S6). Although many research papers report liquid filaments that are formed at Weber numbers (We = $\rho_i U^2 d_i / \gamma$, where $\rho_i$, $d_i$, U and $\gamma$ are the density of the aqueous phase, inner diameter of the needle, injection speed, and interfacial tension, respectively) close to unity[32,36–38], we report the stabilization of liquid columns at We $\gtrsim O(0.001)$ (Figs. 1d and S12), indicating that inertia is not the dominant force that overcomes the surface-tension force in our experiments.

Previous studies report two pathways for the formation of stable liquid columns in a liquid medium: (i) jamming of nanoparticle and end-functionalized polymer entities at the oil-

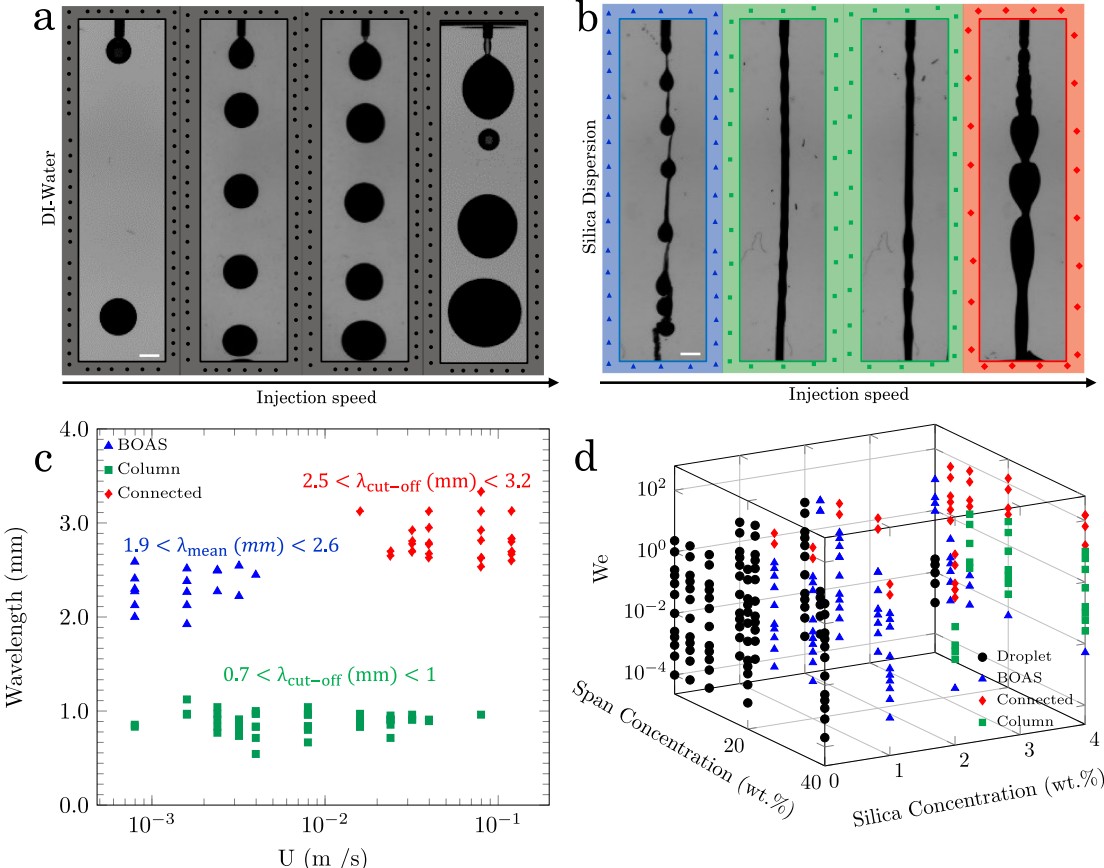

**Fig. 1 Flow regime morphologies. a** DI water and **b** 4.0 wt.% silica dispersion are injected into a 20.0 wt.% Span micellar solution. The scale bar is 1 mm. **c** A plot of the wavelength versus the flow rates identifies four morphologies as single droplet (black), BOAS (blue), liquid column (green), and connected droplet (red), as characterized based on a Fast Fourier Transform (FFT) analysis (Supplementary Notes 2 and 3). **d** Flow regime map of tested Span and silica concentrations at various injection flow rates, where the latter is represented using the Weber number. The map indicates the formation of liquid columns at $We = O\ (0.01)$. Source data are provided as a Source Data file.

water interface[3,6] and (ii) a change of the reverse micelle morphology from spherical to lamella at the oil–water interface[32,39]. In the former, dispersed nanoparticles in the aqueous phase interact with end-functionalized polymers in the oil phase, forming surface-active nanoparticles that are locked at the interface and prevent or delay the Rayleigh-Plateau instability[3,6]. In the latter, lamella structures are produced from the interfacial reaction of dissolved spherical cationic reverse micelles in the aqueous phase and a fatty acid oil phase[32,39]. However, we observe that liquid columns are only formed in Span micellar solutions above 10.0 wt.% (Fig. 1 and Figs. S3–S6). Thus, the surface-activation of silica nanoparticles is not the responsible mechanism for the formation of liquid columns as silica nanoparticles can be surface-activated at much lower Span concentrations (below 1.0 wt.%)[40,41]. To investigate the possibility of the second mechanism, i.e., formation of lamella structures with surfactant assemblies in the presence of silica nanoparticles, we visualize the structure of the interfacial layer with and without silica particles as presented in the next section (more details in Supplementary Note 4, Figs S13–S18).

**Effect of nanoparticles on the formation of liquid columns**. A droplet (volume $V = 2\ \mu l$) of the aqueous phase is placed in the bulk of a 20.0 wt.% Span micellar solution, close to a transparent solid surface (Fig. S18a). Images are recorded over 24 hours from the bottom of the container, as shown in Fig. 2a, b. Initially, a

dark zone is formed at the aqueous phase-oil interface, with an intensity that decreases over time on the DI water droplet (Fig. 2a, 1–6), while it remains constant on the silica droplet (Fig. 2a, 1′–6′). Cryo-SEM images are taken after 24 h from the water-Span micellar solution interface, as shown in Fig. 2a, panels 7–8, which reveal the spontaneous formation of an emulsion zone in which submicrometer water droplets (light gray) are dispersed in the oil phase (dark gray). The droplets are Span reverse micelles that are filled with water after being in contact with the aqueous phase, hence forming the emulsion phase.

In the presence of silica nanoparticles (Fig. 2a, 7′), the droplets in the emulsion phase are more concentrated as compared with the case without silica. However, the droplets in both systems (with and without silica) are spherical. The size of the emulsion droplets, obtained from Cryo-SEM images (Fig. S15), is reduced upon adding silica particles and increasing the particle concentration. The average droplet size changes from ~5 μm for DI water to ~1 μm for 4.0 wt.% silica dispersion (Fig. S16), indicating the enhanced emulsification properties of the system. Inspection of the silica-Span micellar solution interface reveals that the sample from the silica dispersion has a different structure than the one formed from DI water (Fig. 2a, 8′). In addition to the generated micro-scale droplets, we notice the coexistence of interconnected zones of oil and emulsion-rich phases throughout the sample (Fig. S18b). Thus, the silica nanoparticle dispersion generates an emulsion-based interfacial phase that can be a new mechanism for generating stable liquid columns. The formation

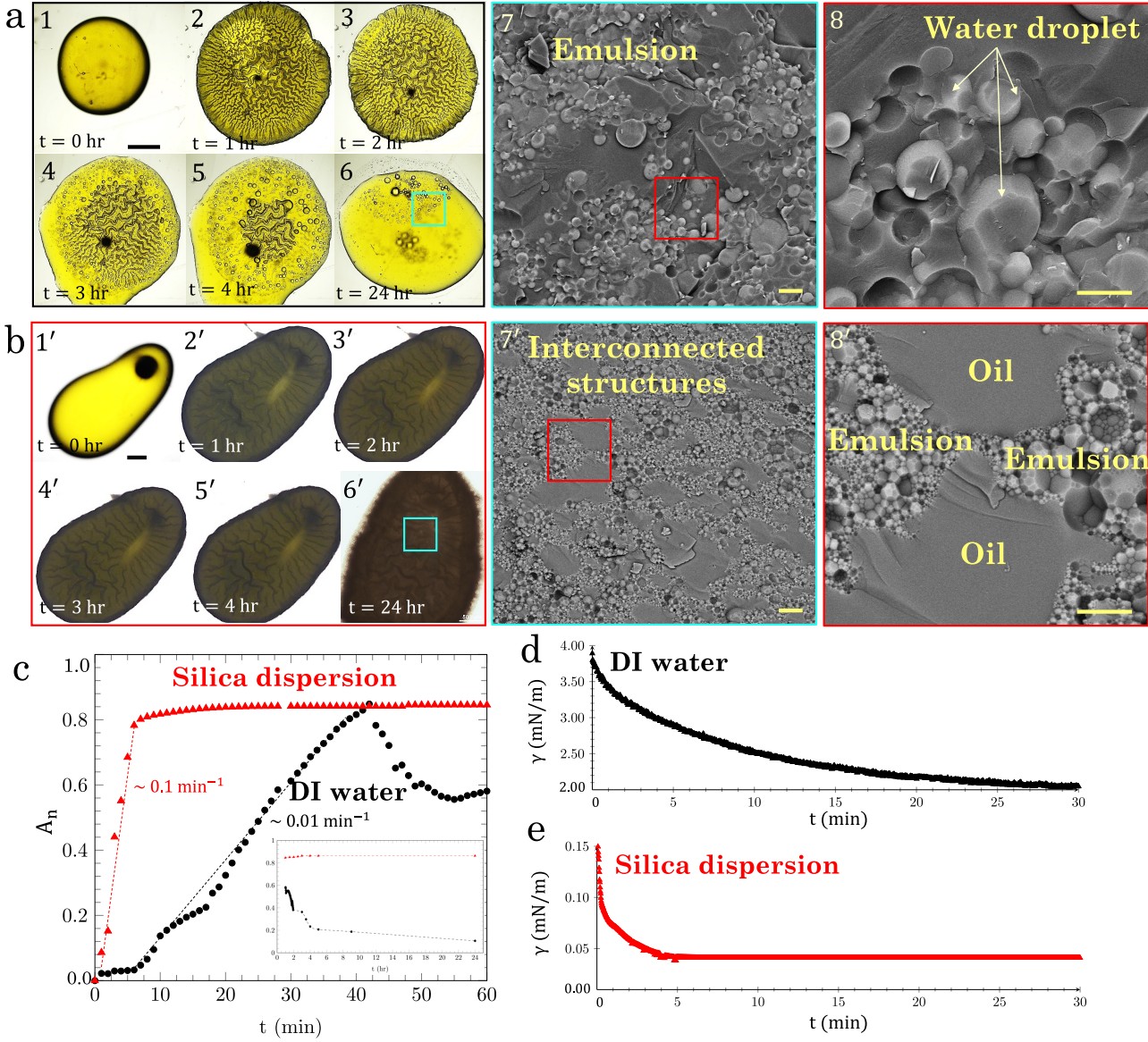

**Fig. 2 Spontaneous formation of interfacial materials.** Bottom view images of the changes over time for a **a** DI water droplet and **b** 4.0 wt.% silica dispersion droplet in 20.0 wt.% Span micellar solution (scale bars = 500 µm). Cryo-SEM images of the interfacial materials generated from DI water and a 4.0 wt.% silica nanoparticle dispersion after 24 hours are shown in (**a**, 7–8) and (**b**, 7′–8′). In the sample generated from DI water (**a**, 7′–8′), an emulsion zone containing submicrometer water droplets is formed. In the sample formed from the silica dispersion (**b**, 7–8), interconnected structures of oil and emulsion phases are formed uniformly throughout the sample. In the cryo-SEM images, the dark and light gray colors represent, respectively, oil and aqueous phases, while the nanoparticles are not visible. The scale bar in (**b**, 7 and 7′) is 10 µm and in (**b**, 8 and 8′) is 5 µm. **c** Rate of emulsion phase formation. The surface area covered by emulsions divided by the total surface area of the droplet is defined as $A_n$ and plotted over 60 min. The data points of the DI water droplet and 4.0 wt.% silica dispersion droplet are shown in black and red, respectively. The slope of initial regime in each case represents the rate of emulsion phase formation. The inset of the figure shows $A_n$ over 24 h. Dynamic interfacial tension ($\gamma$) of **d** DI water-20.0 wt.% Span micellar solution and **e** 4.0 wt.% silica dispersion-20.0 wt.% Span micellar solution. $\gamma$ of DI water decreases slowly over time while $\gamma$ of the silica dispersion remains constant after 2 min. Source data are provided as a Source Data file.

of the oil-emulsion interconnected phase, and consequently liquid columns, is also observed at a lower Span concentration of 10.0 wt.% (Supplementary Note 4).

The time evolution of the dark zone at the surface of the aqueous droplet, normalized by the droplet surface area, and denoted ($A_n$), at each time step is plotted in Fig. 2c (black for DI water and red for a 4.0 wt.% silica dispersion). Since the overall surface area of the droplets is quantified in Fig. 2c, the drop size polydispersity evaluated from Cryo-SEM images (Fig. S16) is not detectable in the results obtained from color camera images. At the interface of the DI water droplet, the fractional change in the

area of the dark zone is generated at a rate of ~0.01 min$^{-1}$ over 40 min, after which it decreases. The inset panel in Fig. 2c shows $A_n$ from 1 to 24 h, where it decreases to less than 0.1 for the case of DI water. The rate of emulsion phase formation is ~0.1 min$^{-1}$ at silica dispersion-micellar solution interfaces, and reaches a plateau after ~4 min. The stability of the emulsion-based structures generated from a nanoparticle dispersion might be due to the presence of nanoparticles at the oil-water interface. Our rheology experiments confirm the formation of viscoelastic interfacial materials at the silica dispersion-span micellar solution interfaces (Supplementary Note 5, Fig. S19).

The interfacial tension ($\gamma$) of DI water and 20.0 wt.% Span micellar solution is ~4 mN/m initially, and decreases to ~2 mN/m after 30 min. In contrast, for a 4.0 wt.% silica dispersion and 20.0 wt.% Span micellar solution, $\gamma$ is initially one order of magnitude lower (~0.16 mN/m) than that of DI water and reaches equilibrium (~0.08 mN/m) after about 2 min (Fig. 2d, e). The low values of interfacial tension of the silica dispersion can be the driving force for the rapid interfacial expansion and the formation of emulsion-based structures. Emulsions are generated by two mechanisms in the presence of silica particles: (i) the presence of reverse surfactant micelles that intake the aqueous phase and (ii) the low interfacial tension. Under very low interfacial tension conditions, the interfacial area would sponta-neously increase by forming small droplets close to the interface. In the case of high viscosity oil, the interfacial area is extended by the formation of droplets and folding, resulting in the penetration of fingers from one phase into the second phase[42]. Later, the invaded streams turn into small droplets. The expansion of interfacial area by the penetration of emulsions from the interface to the oil phase is shown in Fig. S14 and Supplementary Movie 5.

The formation of interconnected structures in particle-laden systems has been reported through spinodal decomposition[43] or emulsion phase inversion[44–46]. In spinodal decomposition, a single-phase liquid undergoes de-mixing, generating a multiphase mixture. Particles segregate to the interfaces and stabilize the mixture in an out-of-equilibrium state[43,47]. Interconnected structures can also be generated through temperature-[48] or concentration-[44–46] driven emulsion phase change. The hydro-philicity of thermo-responsive particle is decreased upon increasing the temperature, hence, the initially particle-stabilized oil-in-water (O/W) emulsion turns into a water-in-oil (W/O) emulsion. Bicontinuous or multiple emulsions are formed at temperatures close to the inversion point[48]. The phase inversion can also occur for partially hydrophobic silica nanoparticles that are initially dissolved in the oil through phase inversion at high particle concentrations (above 1.0 wt.%)[44–46]. Ludox HS 40 silica particles are not thermo-responsive and particle concentration is constant in each test. Thus, the generated interconnected emulsion-based zones in our experi-ments differ from structures with particles reported in the literature.

**Universality of in situ emulsification triggered liquid columns.** Further experiments over a range of micellar solution viscosities from $\mu = 30$ to 1000 mPa s reveal that liquid columns are formed in silica-Span systems with $\gamma_{eq} = O(0.01)$ mN/m and $\mu > 60$ mPa s (Supplementary Notes 6 and 7, Figs. S20–S22). To unravel the general conditions for the generation of liquid columns, we analyze different times scales of the process. The residence time, $t_R = L/U$, is defined as the time elapsed from the injection to the time the injected fluid reaches the bottom of the container, where U is the injection speed and L is the distance between the nozzle tip and the container base. The filament breakup can be pre-vented if the surface-tension-driven Rayleigh–Plateau instability is suppressed within the residence time. The time scale of the Rayleigh–Plateau instability depends on the viscosity ratio of the inner to the outer fluid and the surface tension of the system[49–53]. Decreasing the interfacial tension and increasing the viscosity of the liquid phases reduce the disturbance growth rate and delay the development of the instability in the form of thread breakup and droplet formation.

We calculated the perturbation growth rates and estimated the breakup time for two immiscible liquids from the prediction of the Rayleigh–Plateau instability theory[50,52,54] (Supplementary Note 7). For the case of clean water (without nanoparticles), the experiments

show droplet formation (i.e., an unstable thread) in all tested Span 80 concentrations and injection flow rates (Fig. S3). The estimated time from the Rayleigh-Plateau instability theory is ~0.1 s, which has the same order of magnitude as the droplet formation (Fig. S3). Hence, the experiments support the theory for clean water-Span solutions. Adding nanoparticles to water considerably reduces the effective interfacial tension, which consequently increases the estimated time for the droplet breakup to ~1 s for the two higher silica particle concentrations (2.0 and 4.0 wt.%). So, for silica dispersion-Span solution systems, one expects that the droplet formation occurs for a residence time greater than one second. However, experiments show that the silica dispersion thread remains stable without droplet formation in silica 2.0 wt.%-Span 40.0 wt.% and silica 4.0 wt.%-Span 10.0–40.0 wt.% (Figs. S5–S6) for times much greater than 1 s. Therefore, other mechanisms, besides the low interfacial tension of silica dispersion-span micellar solution and the high viscosity of the outer fluid, could contribute to the attenuation of Rayleigh–Plateau instability within the time frame of experiments.

The rapid in-situ formation of emulsions at silica dispersion-Span 80 micellar solution interfaces, as described in Fig. 2, may suppress the Rayleigh–Plateau instability in our systems. As revealed by experiments (Figs. S5–S6), liquid columns are formed at an intermediate range of flow rates (residence time). For shorter time scales, emulsions do not have enough time to be generated at the interface, thus, for such small residence times, the emulsion layer has not been formed fully to cover the oil-water interface and consequently cannot prevent the Rayleigh–Plateau instability. We refer to the smallest residence time in which liquid columns can be generated as the emulsification time ($t_E$). Despite the fact that emulsions have enough time for their formation at the longest residence times, we cannot see the formation of stabilized liquid filaments at these residence times either. Thus, aside from emulsification, another mechanism, i.e., emulsion drop removal from the interface, is involved, which affects the stabilization of liquid filaments. Since we do not have any external source of flow, the main mechanism for the detachment of emulsions from the interface can be the diffusion of swollen reverse micelles (emulsion droplets) from the interface into the bulk. The longest residence time in which the liquid columns are formed is considered as the diffusion time ($t_D$). In summary, in our system, the Rayleigh–Plateau instability is attenuated by the in-situ formation of an emulsion layer at the aqueous phase-oil interface.

We hypothesize that liquid columns are generated when $t_E < t_R < t_D$. This criterion, as indicated schematically in Fig. 3a, suggests that the liquid columns are formed when emulsions are generated within the time frame of an experiment, but do not diffuse from the oil–water interface into the bulk oil. To characterize the dynamics and column stabilization, we define two dimensionless parameters, representing the relative impor-tance of residence time and diffusion time scales to that of emulsification, as the convection Damköhler ($Da_C = t_R/t_E$) and diffusion Damköhler ($Da_D = t_D/t_E$) numbers.

First, we estimate the emulsification and diffusion times for a set of experiments where flow regime transitions occur, i.e., BOAS to column and column to connected, by increasing the injection speed (Fig. S22). The BOAS regime appears at the longest residence times, meaning that emulsions are generated in this regime and diffuse from the interface (Fig. 3a, 1). Thus, the transition from the BOAS to the column regime is assumed to estimate the diffusion time. The connected regime emerges at the shortest residence times, where emulsions are not formed within the time frame of the experiment (Fig. 3a, 3). Hence, the transition from column to connected is defined as the emulsification time (Table S1).

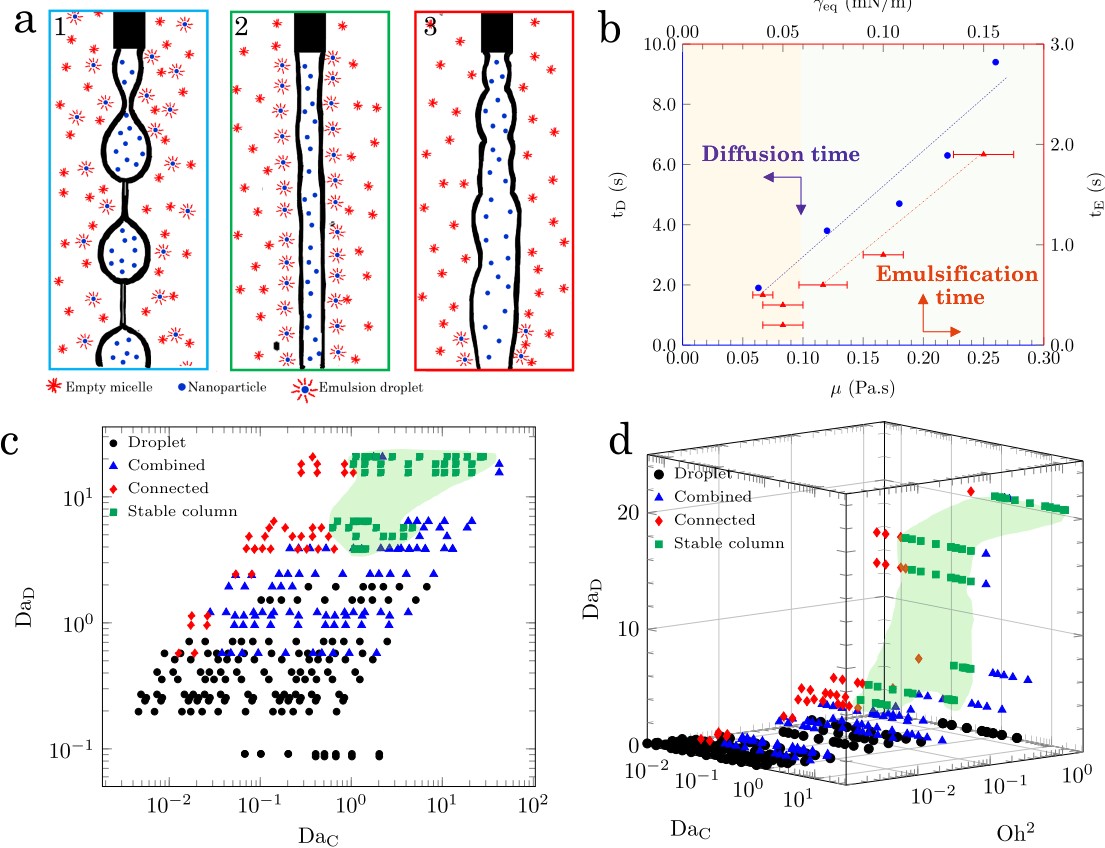

**Fig. 3 Dimensional analysis of the flow regimes. a** Schematic of the flow regime transition from BOAS to column and connected. **a**.1 The residence time is longer than the diffusion time. Emulsions are formed at the interface and diffuse to the bulk. **a**.2 The residence time is longer than the emulsification time and shorter than the diffusion time. Thus, emulsions are generated and remain at the interface. **a** The residence time is shorter than the emulsification time, so that there is not enough emulsion at the interface to stabilize the liquid filaments. **b** Diffusion time, $t_D$ (left and bottom axis in blue), and emulsification time, $t_E$ (right and top axis in red), obtained from the flow regime transitions in (Fig. S22, Table S1). The slope of lines are 31.5 m s²/kg and 11.2 s³/g, respectively. The error bars represent the standard deviation of interfacial tension values of three systems that have a similar emulsification time scale. The yellow and green shaded zones represent regions where the emulsification time is constant and changes upon increasing the interfacial tension, respectively. **c** Relative importance of diffusion and emulsification time scales as depicted by the diffusion (Da$_D$) and convection (Da$_C$) Damköhler numbers. **d** 3D phase diagram of the flow regimes by considering the properties of the inner fluid. The highlighted zone in green shows the region of liquid column formation. Source data are provided as a Source Data file.

The experimentally defined diffusion times in Table S1 increase upon increasing the micellar solution viscosity (Fig. 3b, left and bottom axes in blue). This response is consistent with the Stokes–Einstein theory for diffusion in solution, where the diffusion time is a linear function of the medium viscosity $(t_D \sim \mu)$[55,56] (Supplementary Note 7).

A plot of emulsification time in Fig. 3b (right and top axes in red) shows that in systems with $\gamma_{eq} < 0.06$ mN/m, the emulsification time is 0.2–0.5 s, independent of the interfacial tension value. In contrast, in systems with $\gamma_{eq} \geq 0.06$ mN/m, the emulsification time is a linear function of $\gamma_{eq}$. The equilibrium interfacial tension value is the key factor that controls the spontaneous emulsification process, driving the system toward equilibrium[42]. Hence, the emulsification time can be estimated as $t_E \sim \gamma_{eq}$ (Supplementary Note 7).

The convection and diffusion Damköhler numbers are calculated for a wide range of silica and Span concentrations and inner and outer fluid viscosities (Figs. S3–S6, S20, and S22) based on the time scale analysis in Fig. 3b. A plot of Da$_C$ versus Da$_D$ in Fig. 3c reveals that stable liquid columns are generated for $0.5 \leq \text{Da}_C < \text{Da}_D$, which verifies the hypothesized criterion. The coupling effect of emulsification kinetics (Da$_C$, Da$_D$) and hydrodynamics on the flow regimes are plotted in a three-dimensional phase diagram of

(Da$_C$ – Da$_D$ – Oh²) in Fig. 3d (details in Fig. S23), where the Ohnesorge number, $\text{Oh} = \mu_i/\sqrt{\rho_i \gamma d_i}$, is a material property-related nondimensional number representing the viscous to inertia and surface-tension forces and is calculated based on the inner phase viscosity ($\mu_i$) and density ($\rho_i$), and $d_i$ is the inner diameter of the needle. The results in Fig. 3d categorize the four flow regimes observed in Fig. 1 where the zone of stable liquid columns is highlighted in green.

**Liquid column characteristics and applications**. The stability of liquid filaments during continuous injection was shown in Fig. 1. We also test their stability and durability after the injection has ceased. The 4.0 wt.% silica dispersion is injected in a 20.0 wt.% micellar solution forming a stable column. Once the injection is stopped, the liquid inside the column drains immediately leaving a wrinkled tube skin. The generated emulsion phase forms a viscoelastic skin keeping the structure of the column intact. Upon re-injecting the liquid, the column is restored to its original shape (Fig. 4a and Supplementary Movie 6). The cycle of injection then stopping flow is repeated three times and can be further continued without observing any thread deterioration.

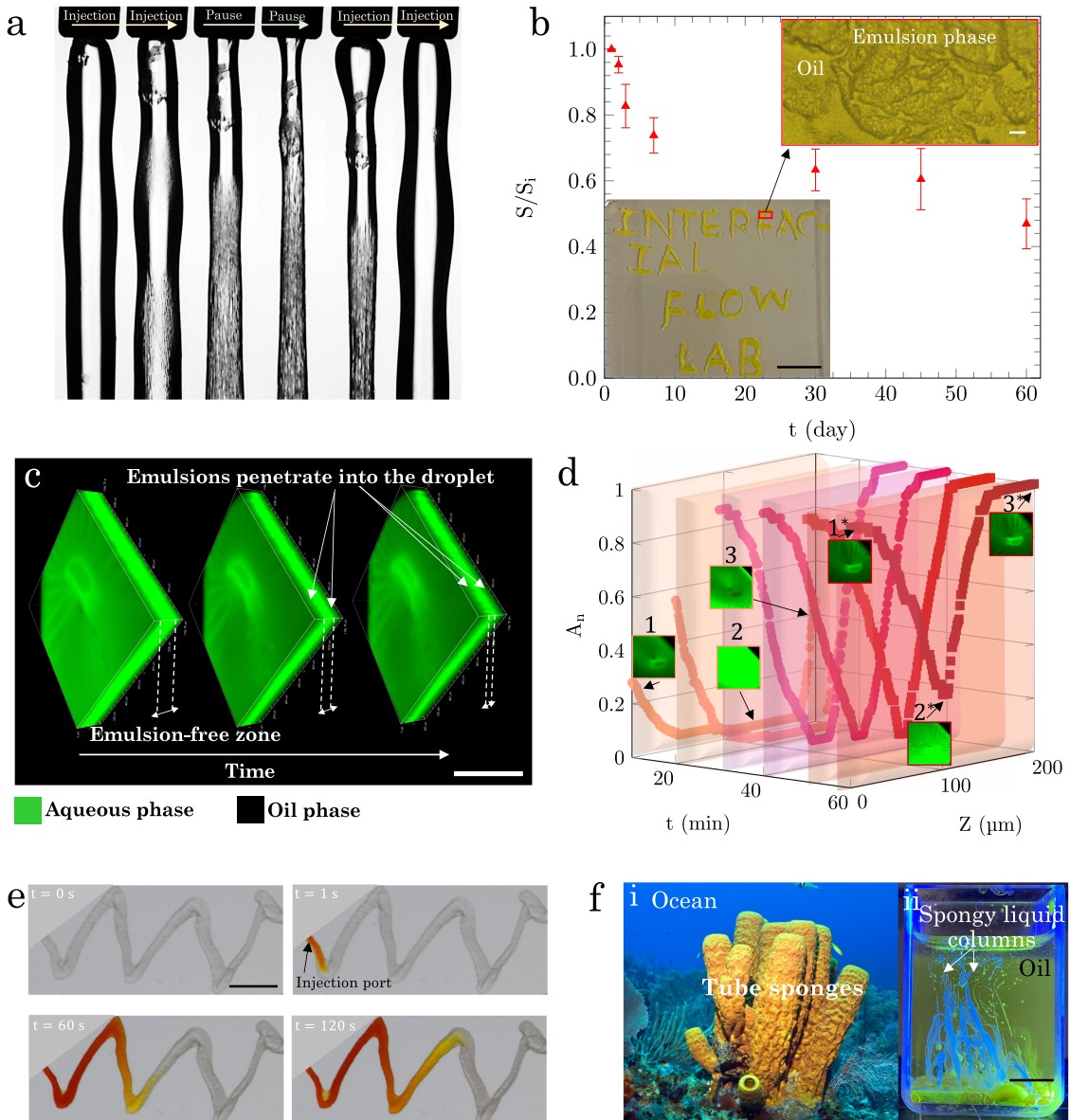

**Fig. 4 Applications of stable liquid filaments. a** The stability of a liquid column during the injection-pause cycles. The outer diameter of the needle is 500 μm. **b** Printing liquid letters and their stability over two months. The surface area of each letter (S) is divided by its initial surface area ($S_i$) and plotted as a function of time. The error bars show the average change in surface area of 20 printed letters. The inset at the bottom shows the printed letters (scale bar is 1 mm) and the one at the top indicates the formation of interconnected structures on the surface of the letters (scale bar is 10 μm). **c** 3D confocal images that show the formation of emulsions and their advancement in depth over time for 4.0 wt.% silica dispersion droplet in 20.0 wt.% Span micellar solution. Green and back colors represent silica dispersion and oil phases, respectively. The shade of green-black color indicates formation of emulsions. The scale bar is 500 μm. **d** Confocal images are captured from the bottom ($z = 0$) to the top ($z = 200 \mu m$) of the droplet with a resolution of 1 μm in depth and 2 μm in the x–y plane at 10 min time intervals. Axes show time intervals, location in z, and surface coverage of the emulsions. The data points at each time interval are represented with a different color to aid the visualization. **e** Injection of DI water (yellow) into the hand-printed silica dispersion channel. **f** Similarities between (i) tube sponges in oceans and (ii) spongy liquid columns. The image in (i) is taken from iStock.com/Kevin Panizza. The yellow tube sponge is relatively large and it can grow up to 1 m in length. The scale bar in (ii) represents 1 cm. Source data are provided as a Source Data file.

Moreover, the silica dispersion is loaded into a micro-pipette (volume 10 μl) and liquid letters with sharp and rounded edges are manually printed in the micellar solution (Fig. 4b). A high-resolution microscopic image of a printed letter shows the presence of the interconnected zones of oil and emulsion at the surface of the letter, which maintains the integrity of the structure. The long-term stability of the printed structures, calculated by measuring the surface area of each printed texture (S) divided by its initial surface area ($S_i$), is plotted in Fig. 4b. The

printed structures remain stable up to 2 months without losing their sharpness (Fig. S24).

Next, a slice (1/4) of a silica dispersion droplet at three different times is shown in Fig. 4c. Initially, the mixed green-black zones (emulsion phase) are only formed at the top and bottom of the droplet, which are in direct contact with the micellar solution. Over time the thickness of the emulsion phase increases, indicating that the emulsification advances into the depth of the droplet.

We also analyze the penetration depth of the emulsion-based structures inside the printed textures. The internal structures of the aqueous droplets previously shown in Fig. 2a, b are captured using confocal microscopy by scanning from the bottom to the top of the droplet in 1 μm steps (Figs. S25–S27). Figure 4d illustrates the area covered by the emulsion inside the silica droplet over 60 min at six time intervals. The inset images labeled as 1, 2, and 3 show the selected z stack at $z = 0$, 100, and 200 μm at the first time interval. The asterisk (★) represents the selected images at the last time interval (50–60 min). The intensity of the emulsion in the center increases from 0 to 0.4 in 60 min, indicating the emulsification penetrates into the silica droplet. The penetration length of the emulsions is proportional to the square root of time suggesting that the emulsion penetration is a diffusion-dominated process (Fig. S28, right axis in blue).

The spatiotemporal 4D images of a silica drop deposited in the micellar solution provide information about the depth and rate at which the emulsion phase advances inside the silica drop and verify the formation of a spongy texture. Thus, a silica dispersion can be used as an ideal ink to create all-in-liquid materials that are emulsified over time.

Finally, a tortuous channel of silica dispersion is printed manually (the same method used for printing liquid letters) in a Span micellar solution. Then, the printed channel is punched by the tip of a second micro-pipette and 10 μl of dyed DI water (yellow) is injected (Fig. 4e and Supplementary Movie 7). The injected aqueous phase follows the pathway of the printed texture without disintegrating the channel's structure and the viscoelastic interfacial layer. Also, note that the printed emulsions are continuously rejuvenated if a part of the interfacial layer is ruptured or damaged. Hence, the present approach for creating all-in-liquid materials offers a new paradigm for in-liquid emulsion printing and liquid-based lab-on-chip techniques. We also note the similarity, although different in length scales, between tube sponges in the oceans and spongy liquid columns in Fig. 4f.

## Discussion

In summary, we have demonstrated a strategy to print, all-in-liquid, materials by rapidly creating an emulsion zone that is locked at the oil-water interface. Distinct from previous approaches that rely on nanoparticle-surfactant jamming, which cannot create an internal structure, the discovery of such an in-situ emulsification mechanism dramatically upgrades our capability to print liquids consisting of many small droplets in a single step. The silica dispersion filaments are stabilized at Weber numbers two orders of magnitude smaller than unity due to the rapid formation of emulsions at the interface and the slow diffusion of emulsions from the interface to the bulk phase. The generality of our approach will open a new avenue for a wide range of applications from energy storage[7,57], due to their highly interconnected structures that can be created spontaneously in large volumes with minimal input energy, high surface-to-volume ratios, and the arrest of silica nanoparticles at the extended emulsion droplet surfaces, to liquid-based lab-on-a-chip devices[1].

Previous research on the formation of liquid filaments for 3D printing in micellar solutions is limited to the ionic surfactant-fatty acid systems. A gel phase is generated at the interface by a change in the morphology of reverse micelles from spherical to lamella, stabilizing liquid filaments of surfactant solutions in oleic acid[32,39], submicrometer domains are generated by photopolymerization[31]. The developed approach in the present work differs from studies on the ionic surfactant-fatty acid systems as the stable liquid columns are generated due to the in-situ formation of emulsion-based interfacial layers. Besides, internal

structures are made of emulsion droplets and are generated spontaneously. The silica-doped interfacial materials attain unique features where (i) a column with viscoelastic walls made of emulsions is formed and (ii) the printed structures in the oil phase exhibit long-term stability on the order of months. The proposed technique offers great flexibility for encapsulating different hydrophobic and hydrophilic cargoes inside the printed structure and the surrounding medium, enabling the functionalization of a printed network of droplets for tissue engineering[12] and addressing the challenge of delivery of incompatible materials in biomedical applications[58–60]. It also creates droplet-based structures for a wide range of applications that require interactions between a printed structure and a liquid medium.

## Methods

**Materials**. A 40 wt.% Ludox silica dispersion (HS 40, Sigma) is used as the concentrated source of nanoparticles. Silica nanoparticles in the dispersion are spherical with an average diameter of 7.9 ± 0.1 nm. The silica dispersion contains sodium counter ion with the concentration of <0.08%[61]. Sorbitan monooleate (Span 80, Sigma) is used as an oil soluble non-ionic surfactant. Span 80 has molecular weight of 428.60 g/mol and the hydrophilic–lipophilic balance (HLB) of 4.3. We use a sample of mineral oil (Drakeol 35, $\rho = 0.876$ g/cm$^3$, $\mu = 135$ mPa s) as the oil phase. For a few sets of experiments, minerals oils with different viscosities $\mu = 30$ mPa s and $\mu = 1000$ mPa.s are used. A water-based fluorescent color (Cole-Parmer, UZ-00298-19) is used to stain the aqueous phase. All products are used as received without further purification. Unless otherwise stated, the HS silica dispersion, Drakeol mineral oil ($\mu = 135$ mPa s), Span 80, and Span 80 dissolved in Drakeol mineral oil are referred to, respectively, as silica dispersion, oil phase, Span, and micellar solution in the entire manuscript.

**Silica dispersion preparation**. DI water (Direct-Q, Millipore Sigma) is added to the concentrated silica dispersion to reach the desired concentrations of 1.0 ($\rho_i = 1.003$ g/cm$^3$, $\mu_i = 1.0$ mPa s), 2.0 ($\rho_i = 1.009$ g/cm$^3$, $\mu_i = 1.1$ mPa s), and 4.0 ($\rho_i = 1.027$ g/cm$^3$, $\mu_i = 1.2$ mPa s) wt.%. Samples are homogenized in an ultra sonic bath (Isonic P4830) for 10 min.

**Micellar solution preparation**. Span is dissolved in mineral oil at concentrations of 1.0, 5.0, 10.0, 20.0, and 40.0 wt.%. Samples are sonicated in the bath for 30 min.

**Experimental set-up**. Liquid columns are generated in a vertical rectangular cell (2 cm × 4 cm × 1 cm) through a needle with the outer diameter of 500 μm and the inner diameter of 400 μm.

**Spinning drop tensiometer (SDT)**. The dynamic interfacial tensions (IFT) of the nanoparticle dispersions—micellar solutions are measured using a spinning drop tensiometer (SDT, Kruss). The glass capillary is filled with the aqueous phase and an oil drop is placed at the top of the plastic cap. The spinning rate is in the range of 6000–8000 rpm, and data are recorded for 2000 seconds. Vonnegut's equations is used to obtain the interfacial tension values for each sample. $\gamma = (\rho_{\text{aqueous phase}} - \rho_{\text{micellar solution}}) \omega^2 R^3/4$, where $\omega$ and $R$ are, respectively, the spinning rate and the radius of the liquid cylinder.

**Density measurement**. The densities of the aqueous phases (silica dispersions) are measured using Density2Go (Mettler Toledo) and the densities of the oil phases (micellar solutions) are measured by DMA 35 (Anton Paar).

**Viscosity measurements**. The viscosities of the aqueous phases (silica dispersions), micellar solutions, and emulsions are measured at shear rates of $10^{-3}$–$10^3$ (1/s) using Discovery Hybrid Rheometer (DHR-2, TA Instrument). The rheological properties of the samples are measured in a cone and plate geometry with a cone diameter of 20 mm and angle of 2°.

**Confocal laser scanning microscopy (CLSM)**. Images are captured with 4× (CFI Plan Apo Lambda, Nikon, resolution of 6.2 μm/pixel) and 10× (CFI Plan Flour, Nikon, resolution of 2.3 μm/pixel) magnifications using confocal laser scanning microscopy (Nikon A1R). The laser power is set at 10–15 mW, and the pinhole is fixed at 1.2 AU for all images. We use 488 nm (FITC) and 561 nm (TRITC) laser emissions to detect the oil phase and silica dispersion, respectively.

**Cryogenic-scanning electron microscopy (Cryo-SEM)**. A variable pressure environmental field emission scanning electron microscope (FEI Quanta 250 FEG) coupled with a Gatan Alto2500 cryo-transfer system is used to capture images of the emulsions. Samples are freeze-dried in liquid nitrogen before the imaging.

**Bottle experiments**. Three milliliters of an aqueous phase (DI water or silica nanoparticle dispersions) is poured into a glass vial 20 ml) and 3 ml of an oil phase is gently added at the top of the aqueous phase using a micro-pipette. Emulsification starts spontaneously without shaking, stirring, or homogenizing.

**Image analysis**. Image Fiji software is used to analyze the color camera and confocal images and calculate porosity of 3D structures. The droplet size is measured using Analyze Particle modulus. 3D images are imported as a sequence of z-stack images and reconstructed using the 3D Objects Counter modulus. Then, the porosity is calculated by measuring the void volume of each image divided by the total volume.

**Fourier transform analysis**. The time series images of the injection of aqueous phase into micellar solutions (Figs. S3–S6) at different concentrations of nanoparticles and surfactants and the injection flow rate are binarized with an appropriate threshold value in image Fiji software. The power spectrum is calculated at each y (location along the column) from the summation of threshold pixels in x direction. The data is obtained from plot profile. Then, the average power spectrum is calculated using 1D Fourier analysis in Matlab.

Note. All experiments are conducted at atmospheric pressure ($88.4 \pm 0.8$ kPa) and temperature ($21 \pm 0.5$ °C). IFT, density, and viscosity experiments are repeated at least three times. Emulsion samples are generated at least ten times to confirm the reproducibility of the spontaneous emulsification. Image analysis is repeated four times for each sample.

## Data availability

Source data are provided with this paper as an excel file. A full series of CLSM images captured can be provided when required. Source data are provided with this paper.

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

## Acknowledgements
We would like to acknowledge Dr. G.M. (Bud) Homsy for numerous fruitful and enlightening discussions and comments during the course of this work. We appreciate iStock.com/Kevin Panizza for sharing Fig. 4f(i). This study was financially funded by Natural Sciences and Engineering Research Council of Canada (NSERC) Discovery Grant RGPIN/07186-2019, University of Calgary's Canada First Research Excellence Fund (CFREF) program, and the Global Research Initiative (GRI) in Sustainable Low Carbon Unconventional Resources. We also gratefully acknowledge infrastructure funding from Canadian Foundation for Innovation (CFI) CFI JELF 33700. P.B. appreciates Alberta Innovates and the University of Calgary Eyes High Graduate Student Scholarships.

## Author contributions
P.B. conceived the main idea for the study, designed and conducted the experiments, and prepared the figures. P.B. and S.H.H. performed the initial result interpretation and wrote the first draft. P.B., H.A.S., and S.H.H. discussed results, contributed ideas for mechanism interpretation, edited on the manuscript, and provided feedback on the final version of the paper.

## Competing interests
The authors declare no competing interests.
