## [Peer Review File · Nature Communications]

Spongy all-in-liquid materials by in-situ formation of emulsions at oil-water interfacesReviewers' comments:

Reviewer #1 (Remarks to the Author):

The authors report a liquid-in-liquid printing process that creates a microemulsion or bicontinuous phase at the interface between the two liquids. There are a growing number of liquid-in-liquid printing processes. Unlike the authors' claim, it is entirely possible to print continuous threads and overcome the Plateau-Rayleigh instability. This cannot be refuted. See, for example, the work of Russell and co-workers on the use of nanoparticle-polymer surfactants, which enable that fabrication process in water-oil and water-water systems. Nevertheless, the creation of a mechanically robust microemulsion/bicontinuous phase at the interface during printing is interesting from a fundamental point of view.

What is perhaps unfortunate is the authors loose language and unsubstantiated claims in their discussion of some of the major outcomes. For example, the authors claim that they create "porous liquids" in their fabrication process, which does not appear supported by their data. In addition, the term "porous liquid" has already been taken, so to speak, by researchers elsewhere (Andy Cooper and others) who create porosity in liquids by using discrete and soluble/dispersible cages whose free volume excludes the solvent on the basis of size. There are many papers on this topic and therefore I am not sure why the authors have chosen this as a terminology for a result they have not provided evidence to support.

In addition, the authors have not explored much in the range of materials classes that might exhibit such behavior. In any sort of revision, if invited by the editor, an expansion in scope would need to be established. It would also be necessary to show with data what new or intriguing properties or functions are available to this class of liquid-in-liquid materials. The field of liquid-in-liquid printing has reached a level of maturity that variations on a theme are not likely to have impact unless the functions of those materials make use of the unique materials hierarchy. If such issues were addressed in a substantial revision, then it might be considered suitable for publication in Nature Communications. However, in its current state, only basic principles of fabrication have been established, and not the breadth of applicability or functionality enabled.

Reviewer #2 (Remarks to the Author):

This paper presents a simple and novel approach to create stable liquid filaments of silica nanoparticle dispersions and propose to use them as inks to construct materials consisting a network of droplets. The phenomena is well characterized and demonstrated. The mechanism of forming the liquid filaments with the presence of silica particles is well explained. The strategy of creating liquid columns by in-situ emulsification is general and should be potentially beneficial to any applications that involve highly interconnected structures with high surface-to volume ratios. Therefore, I suggest to accept this manuscript after addressing the following minor comments.

1. Figure 1d is not readable at all. I suggest to decompose it into several ones to make the point clearer.
2. On page 7, line 84, the author state that "water droplets are Span micelles that are filled with water after being in contact with the aqueous phase". It would be nice if author can provide any evidence or reference to support this point.
3. On page 10, line 118, the author should first explain the definition and the physic meaning of emulsification time and diffusion time?

Reviewer #3 (Remarks to the Author):

The submitted manuscript reports on liquid-in-liquid printing of filaments that are stabilized by emulsions that form when an aqueous solution containing silica nanoparticles come in contact with an oil phase containing Span 80 micelles. Four different flow regime morphologies are reported and include single droplets, bead-on-a-string, column, and connected. The transition from single droplets to the three other morphologies is a result of the addition of silica nanoparticles and increased injection speed. Optical and scanning electron microscopy images indicate that the emulsion morphologies appear different between DI water and the aqueous solution containing silica nanoparticles. The formation of smooth liquid columns is hypothesized to be a result the residence time is between emulsification and diffusion times, and is controlled by the injection speed. A few examples are shown indicating that it is possible to print liquid filaments.

Overall, the liquid-in-liquid printing results are interesting and reveal a new method for printing liquid structures. Although some of the stated claims are supported by the results, there are areas that need to be addressed and revised. The major concern is the classification that the microemulsion morphology is bicontinuous when silica nanoparticles are present. Once the authors address the minor comments below, the manuscript is publishable.

Comments

1. How are the authors defining the bicontinuous morphology? In surfactant and diblock copolymer literature, the bicontinuous microemulsion structure is defined as having zero mean curvature and negative Gaussian curvature. The microemulsion phase in the SEM images look to have non-zero mean curvature. Therefore, it is highly unlikely that a bicontinuous morphology forms.
2. An alternative claim is that the silica nanoparticles preferentially organize at the aqueous/oil interface, creating a more elastic droplet interface that jams the system. Nanoparticles are known to be effective emulsifying agents. It is hard to determine from the SEM images, but it looks like the droplet diameter decreases with added silica nanoparticles. This would help confirm the emulsifying characteristics of the nanoparticles.
3. The rapid decrease in interfacial tension with the addition of nanoparticles suggests that they reside at the aqueous/oil interface. How does the stabilization time of the interfacial tension change of with silica nanoparticle wt%?
4. Page 7, Line 90: There may not be a change in the Span micelle morphology, but the optical and scanning electron microscopy images do not probe nanometer length scales that would be necessary to support the claim. Please revise. The comment is also made in the conclusion section.
5. There is a recent article showing that it is possible to print robust liquid filaments with internal phase nanostructures using both ionic and non-ionic surfactants (Macromol. Rapid Commun. 2021, 42, 2100445). Thus, the statement in the conclusion that 3D printing of micellar solutions is limited to ionic surfactants is not correct. The authors seemed to have missed the reference.

Reviewer #4 (Remarks to the Author):

The comments are attached.

In this manuscript, the authors study the morphological states and flow instabilities of ternary/quaternary mixtures of mineral oil-Span-water (w or w/o Silica nanoparticles) with some applications in printing. There are some concerns regarding the validity of the proposed mechanism and the significance of the work. Also, the connection between fluid dynamic results and direct applicability of them to printing is not discussed. I have following specific comments.

Major comments:

1. The discussion about underlying stabilization phenomena is not convincing and the arguments are not completely correct or well supported. For example, the term microemulsion are misused (e.g., figure 2 and line 166, line 83). In the colloids field, the microemulsion is referred to a thermodynamically-equilibrated phase that possess features in the order of only a few nanometers (such as micellar, inverse micellar) thus not possible to be captured in the micro-meter scale as in figures 2, S12, and S19. There are major differences in terms of stability and scales in emulsion and microemulsion definitions that are not properly addressed throughout the paper. Furthermore, it is claimed that the in situ emulsification could be the underlying mechanism for stabilization of interface, however emulsified phases are often unstable with insufficient mechanical properties (such as water-in-oil droplets or oil-in-water emulsions), unless they have some internal nanostructures, that are not addressed here. It is also relevant to discuss the possible phase changes in emulsification or formation ouzu effects that are commonly used in the well-established area of nanoprecipitation. Also, it is not clear how the authors concluded that the micelle morphology remains unchanged in the presence of absence of nanoparticles (line 90) by looking on the SEM pictures (which has droplets of micrometer sizes), since the micelles (usually in the order of a few nanometers) cannot be captured in that image.

2. It is mentioned that the interfacial layer has elastic/viscoelastic properties in the paper. However, no characterization on interfacial properties has been performed to back it up. The authors are advised to perform necessary characterization such as interfacial rheology at the liquid-liquid interface or bulk rheology on isolated gel-like samples. Furthermore, the terms “elastic” and “viscoelastic” are used interchangeably (in the main text and in the supporting information). These two terms are not clearly the same. Use of interfacial and bulk rheology is needed to determine whether mechanical properties of interfacial materials suggest an elastic or viscoelastic behavior.

3. The manuscript shows interesting observations and exhaustive fluid dynamic data for a system of oil-surfactant-water (w or w/o nanoparticles). However, the advantages of the current system compared to other relevant systems are not well supported. The authors are advised to include convincing arguments concerning the novelty of their work. In doing so, the introduction should be more comprehensive covering current approaches in liquid-in-liquid printing (e.g., [1-2,4] and all various underlying mechanisms as well as a discussion section covering comparisons of their systems with others. The author mentioned a few comparisons somewhere else in the paper (jamming, phase transition, section 3.1) or SI (emulsion-based in table s2), but not discussed it thoroughly in the introduction and the discussion. Also, some of the advantages that claimed about the system are not convincing. For example, in table S2 (and throughout the main text) it is claimed that the system is capable of encapsulating cargos and method 1 and 2 are not. However, they have not performed any encapsulation test to support that. Also, the stability that claimed as the strength of their work (in table B2 and main text line 169) are not supported or at least not illustrated in Figure S18. Structures seem they have lost their shape (such as letter F, L). Another claim about the internal structure and porosity of the structure has

not been tested (e.g., for a good porous example:[3]). It is not clear what length scale they are talking about for the internal structures. For example, they mentioned that the internal structure is not available for the method 1 in Table S2, however, the recent publication from that group supports the formation of internal nanostructure [4].

1. Forth, J., Liu, X., Hasnain, J., Toor, A., Miszta, K., Shi, S., ... & Russell, T. P. (2018). Reconfigurable printed liquids. *Advanced Materials*, 30(16), 1707603.
2. Lin, D., Liu, T., Yuan, Q., Yang, H., Ma, H., Shi, S., ... & Russell, T. P. (2020). Stabilizing Aqueous Three-Dimensional Printed Constructs Using Chitosan-Cellulose Nanocrystal Assemblies. *ACS applied materials & interfaces*, 12(49), 55426-55433.
3. Sears, N. A., Wilems, T. S., Gold, K. A., Lan, Z., Cereceres, S. N., Dhavalikar, P. S., ... & Cosgriff-Hernandez, E. M. (2019). Hydrocolloid inks for 3D printing of porous hydrogels. *Advanced Materials Technologies*, 4(2), 1800343.
4. Honaryar, H., LaNasa, J. A., Lloyd, E. C., Hickey, R. J., & Niroobakhsh, Z. (2021). Fabricating Robust Constructs with Internal Phase Nanostructures via Liquid-in-Liquid 3D Printing. *Macromolecular Rapid Communications*, 42(22), 2100445.

4. The description of initial oil phases and their interpretation are not clear. For example, 20 wt% span solution (in mineral oil) is claimed to be a micellar solution, however, since the continuous phase is an oil, one can expect the formation of reversed micelles and not micelles. Also, the rheological properties of that phase suggested a shear thinning behavior (Fig S1b), however, it is unlikely that the micellar phases alone can show non-Newtonian behavior. If possible, proper characterization (such as x-ray scattering) should be used to support the internal structures of such phase. Authors need to integrate the structure and properties into understanding the underlying phenomena in the main text discussion.

5. The title should be more descriptive of what is presented in the paper, i.e., include the underlying mechanisms or process. Also, the title includes "printing" which is not investigated at all, as no 3D model was used for printing, rather all the liquid structures were created. Furthermore, it is not clear why the terms "spongy", "all-in-liquid" are used, for instance, spongy texture is not well illustrated in Fig 4f. The spongy structure actually is a well-defined nanostructure in colloids field and use of spongy liquid here could be misleading.

6. The authors assigned emulsification time t_e and diffusion Damkoeler time t_D based on some initial assumption on flow column morphology (as in Section 3.3 and fig 3a), which is questionable. Authors claimed that the time scale t_R should be between emulsification time t_e and diffusion Damkoeler time t_D as the criterion for the formation of the liquid column. The argument to support that criterion is not clear. Also, there is almost no connection between the flow column morphological states observed in section (3.1) with printing application in section (3.3). It is advised to discuss what flow morphologies can be used for printing and what conditions should be considered.

Minor comments:

1. In line 29, the authors reported the average speed for injection, but there is no mention of how they calculate those values.
2. In line 138 specifically and throughout the manuscript in general, authors report that at an interfacial tension higher than 0.6 mN/m, the emulsification time is 0.2-0.5 s. From what we can see based on Figure 3b, we believe they meant 0.06, not 0.6 mN/m.

3. In the “methods and material” section and for the “viscosity measurements”, no information is provided about the rheometer geometry that was used (parallel plate or cone & plate). Also, it is advised to include more details about the test procedure (what shear rate range was used for setting up the test, what gap was used).
4. In the supporting information, the shear rate is calculated in the experiments using “ $8U/di$ ”. Where is this equation is coming from and can authors provide a reference for it?
5. Many pictures (especially SEM pictures in main text and SI, e.g., fig1b7’) are too small. The scale bar either missing or not clearly visible for some pictures. Some captions do not include complete description. For instance, in fig 1-q1-1a6, the scale bar set is not reported. The use of dye for figs 1a and 1b not reported in the caption (it seems that it is yellow-dyed aqueous phase but it can easily get mixed with oil droplets).
6. In Fig. 2 -7,8 droplets look polydisperse, but the surface area (A_n) reported in the plot C does not include any error bar for droplet values. Also, it seems the drops in fig2-1 to 2-6 form some structure, which is not explained in the text or caption.

Reviewer #1 (Remarks to the Author):

The authors report a liquid-in-liquid printing process that creates a microemulsion or bicontinuous phase at the interface between the two liquids. There are a growing number of liquid-in-liquid printing processes. Unlike the authors' claim, it is entirely possible to print continuous threads and overcome the Plateau-Rayleigh instability. This cannot be refuted. See, for example, the work of Russell and co-workers on the use of nanoparticle-polymer surfactants, which enable that fabrication process in water-oil and water-water systems. Nevertheless, the creation of a mechanically robust microemulsion/bicontinuous phase at the interface during printing is interesting from a fundamental point of view.

What is perhaps unfortunate is the authors loose language and unsubstantiated claims in their discussion of some of the major outcomes. For example, the authors claim that they create "porous liquids" in their fabrication process, which does not appear supported by their data. In addition, the term "porous liquid" has already been taken, so to speak, by researchers elsewhere (Andy Cooper and others) who create porosity in liquids by using discrete and soluble/dispersible cages whose free volume excludes the solvent on the basis of size. There are many papers on this topic and therefore I am not sure why the authors have chosen this as a terminology for a result they have not provided evidence to support.

In addition, the authors have not explored much in the range of materials classes that might exhibit such behavior. In any sort of revision, if invited by the editor, an expansion in scope would need to be established. It would also be necessary to show with data what new or intriguing properties or functions are available to this class of liquid-in-liquid materials. The field of liquid-in-liquid printing has reached a level of maturity that variations on a theme are not likely to have impact unless the functions of those materials make use of the unique materials hierarchy. If such issues were addressed in a substantial revision, then it might be considered suitable for publication in Nature Communications. However, in its current state, only basic principles of fabrication have been established, and not the breadth of applicability or functionality enabled.

We appreciate the feedback provided by the reviewer, which has certainly helped us to improve the revised manuscript. We understand that the inappropriate use of the "porous liquid" terminology may have led to some sort of confusion regarding the novelty and application of our work, as we will discuss in this response letter. Next, one-by-one, we provide clarification regarding the concerns raised by the reviewer regarding the field of liquid-liquid printing and the generality of the present technique. We also provide details of the unique features related to the our development of structured liquids based on an interfacial mechanically robust microemulsion/bicontinuous phase at the interface.

Q.1.1 The authors report a liquid-in-liquid printing process that creates a microemulsion or bicontinuous phase at the interface between the two liquids. There are a growing number of liquid-in-liquid printing processes. Unlike the authors' claim, it is entirely possible to print continuous threads and overcome the Plateau-Rayleigh instability. This cannot be refuted. See, for example, the work of Russell and co-workers on the use of nanoparticle-polymer surfactants, which enable that fabrication process in water-oil and water-water systems. Nevertheless, the creation of a mechanically robust microemulsion/bicontinuous phase at the interface during printing is interesting from a fundamental point of view.

R. 1.1 The manuscript demonstrates the formation and application of spongy (stabilized with emulsion layers) liquid columns for utilization in printing liquid letters and creating liquid-fluid channels. We agree with the reviewer that there are growing numbers of recent publications (e.g., already in 2005 Subramanian et al. used particle-covered bubbles for arresting liquid-air interfaces in desired non-equilibrium shapes, which was later extended to liquid-liquid interfaces by **Straford etl al. (2005)** through simulation and by **Herzing et al. (2007)** and **Cui et al. (2013)** through experiments) in the field of liquid-in-liquid printing which indeed show broad interests and great potential for vast applications. However, as stated in many recent publications, this field is still being developed, and there are key challenges and limitations that need to be addressed and resolved, including extending the use of a broader range of materials, creating internal structures within the printed frame, enhancing interactions between the printed structure and the surrounding media. The recent advancement in this field has led to the development of multi-layer all-in-liquid devices by Russel's group [**Lui et al. (2022)**] and creating internal structures by Niroobakhsh's group [**Honaryar et al. (2021)**], however, there is a need for the development of breakthrough techniques [**Cleg et al. (2020)**].

Currently, there are two approaches for liquid-in-liquid printing: (i) nanoparticle-polymer jamming and (ii) change in lamella structures with cationic surfactant solutions and fatty acids (we discuss them in the revised introduction as below). Our contribution is introducing a new technique for printing all-in-liquid materials by forming an emulsion layer in-situ at the oil-water interface. We were pleased to read that the reviewer acknowledged that creating a robust emulsion layer at the interface is an interesting concept. We have developed a new methodology based on this concept for printing a liquid phase in the second immiscible liquid phase. We understand that our introduction in our first submitted manuscript may have brought the impression that we do not have a comprehensive understanding of this field, although we had a summary table mentioning the advantages/shortcoming of the previous works in the supplementary information. To resolve this issue, we added a detailed discussion on the history of liquid-in-liquid printing and the advances of recent works in the introduction (lines 6-35 in the revised manuscript) as:

~~Recently developed~~ Liquid-in-liquid printed materials [6,10-14] have many potential applica-
tions in energy storage [4,5], microreactors [7], and for creating biomimetic materials [8,15].
These types of materials are generated ~~by the jamming of nanoparticles at the oil-water interface~~
by application of an electrical field [16], using molding [11,17], or direct ink writing (DIW) print-
ing techniques [10,12]. Liquid-fluid interfaces can be arrested in desired non-equilibrium shapes
by the adsorption of colloidal particles. For instance, bicontinuous structures, called bijels, are
formed from the spinodal decomposition of a binary liquid mixture containing amphiphilic par-
ticles [18,19]. Spherical bubbles covered with particles [20] and liquid droplets coated with the
assembly of nanoparticles and end-functionalized polymers [16] can be redesigned into various
anisotropic shapes by the particles' jamming at the interface. The driving force for reducing the
surface area jams nanoparticles at the interface and creates a viscoelastic layer that locks the
system in any arbitrary structure. Nanoparticle jamming at the oil-water interface can prevent
jet break-up [10], where stabilized liquid filaments are used to create reconfigurable all-in-liquid
composite materials of oil-water [6,11,12,21-23] and water-water [13,24]. Particle jamming
technique is applicable for a wide range of liquid viscosities and many particle-polymer combi-
nations, and enable the exchange of materials between the printed texture and the surrounding
fluid. However, the generated materials with particle jamming approach lack multiscale poros-
ity created by emulsion-based inks used in 3D printed solid structures [25-27].
Structured liquids with submicrometer domains can be achieved by the solvent transfer
induced phase separation (STRIPS) approach [28,29]. A homogeneous mixture of oil-water-
alcohol-surfactant-particle is rapidly injected into the continuous water phase. The phase sep-
aration starts upon the injection, where the alcohol is extracted into a continuous phase from
the mixture. The presence of particles results in the arrest of the mixture in a transition zone
of bicontinuous structures during the phase separation [29]. Recently, internal structures inside
the all-in-liquid printed materials are accomplished with photocurable polymers [30]. Oil-water
interfaces are stabilized by the interactions of surfactant assemblies, i.e. micelles, with fatty
acids, where micelle morphology is changed from spherical to lamella and a gel phase is formed
at the interface [9,30]. In the presence of photocurable polymers inside the internal phase,
submicrometer regions are formed after photopolymerization [30]. STRIPS approach is limited
to the use of a ternary mixture and photopolymerization eliminates the fluidity of the printed
material. Besides, these approaches do not form an interconnected network of droplets similar
to those with emulsions. Thus, an approach for creating spontaneous and droplet-based 3D
printed all-in-liquid materials, i.e., self-derived compartmental emulsification is required.

Q1.2 What is perhaps unfortunate is the authors loose language and unsubstantiated claims in their discussion of some of the major outcomes. For example, the authors claim that they create "porous liquids" in their fabrication process, which does not appear supported by their data. In addition, the term "porous liquid" has already been taken, so to speak, by researchers elsewhere (Andy Cooper and others) who create porosity in liquids by using discrete and soluble/dispersible cages whose free volume excludes the solvent on the basis of size. There are many papers on this topic and therefore I am not sure why the authors have chosen this as a terminology for a result they have not provided evidence to support.

R.1.2 We agree with the reviewer that the term “porous liquids” has been used loosely and have changed the terminology to “liquids with emulsified interfacial layers”. This language can better represent the interfacial skin made of emulsion drops of silica nanoparticles packed in a continuous oil phase, analogous to the packed beads in synthetic porous media [Colligan et al. (2006), Gueven et al. (2017)]. Hence, the term porous is borrowed from porous media science, where porosity refers to the space between the compacted emulsion drops. We should draw the reviewer’s attention to section IX in the SI (Figures S25-28), where the porosity of the interfacial layer, its

generation and growth have been characterized. We have clarified our writing to make these parts of our contribution more clear and removed the “porous liquid” term throughout the manuscript.

Q. 1.3 In addition, the authors have not explored much in the range of materials classes that might exhibit such behavior.

R.1.3 Studying a wide of range of nanoparticle/surfactant/polymer composites is not feasible as changing materials may significantly alter the fluid dynamics and resulting structured liquids. It should be emphasized that the related published articles in the literature are also focused on one set of nanoparticle-polymer systems. In our work, we have conducted a detailed analysis on the time scales of various competing mechanisms in forming stabilized liquid columns. We have drawn general maps, based on dimensionless groups governing the flow and emulsification dynamics, where the favorable conditions to form structured liquids are distinguished. Such an analysis can be extended to other systems of nanoparticle-surfactant systems. Section X in SI provide a summary of the existing methodologies and the used material in each system, including the present work, and their advantageous and shortcomings (see **Table S2**). We have clarified our writing to make these parts of our contribution more clear.

The advantages of the developed technique can be summarized as: the printed spongy materials (i) are highly interconnected, (ii) can be spontaneously created in large volumes with minimal input energy, and (iii) provide high surface-to-volume ratios with the nanoparticles settled at the extended emulsion droplet surfaces, all unique characteristics of the present study. The printed features can be as small as a millimetre to a few centimeters with the micro-scale domain size. Emulsion drops, generated with the used materials in this work, are submicrometric, as they are packed into distinct layers with micrometer thicknesses. The interfacial skin in the present work is dynamic, meaning that the droplets in the outer layers are slowly detached and diffused into the surrounding phase while a fresh internal layer is generated. The rates of detachment and regeneration of droplets can be controlled or ceased by altering the fluids separated by the skin. As demonstrated in the manuscript, the tube can be readily contracted or expanded, like blood vessels, by the rate of the fluid pumped inside the tube. Moreover, the intensity of droplets can be controlled through the concentration of micelles. Although we have only shown the manual printing of a simple tube-like structure, more sophisticated structures can be printed using a programmable 3D printer.

In summary, liquid-in-liquid printing is an emerging field, with a historical growth of stabilizing liquid-fluid interfaces in desired non-equilibrium shapes from the work of **Straford et al. (2005)**, **Subramanian et al. (2005)**, and **Herzing et al. (2007)** till recent advances related to various features and the use of materials by **Honaryar et al. (2021)** and **Lui et al. (2022)**. We present a fundamental platform for the spontaneous structuring of emulsion drops *in-situ*, which is a missing link in the literature. The approach is built upon our recent report on the spontaneous formation of multiple emulsions, where a new interfacial material with unique layered microemulsion shells is generated. This has been accomplished by engineering the kinetic and hydrodynamic characteristics of the nanoparticle dispersion-micellar solution interfaces. Furthermore, the required criteria for fabricating the *spongy all-in-liquid materials* have been fully unravelled. In previous works, the liquid-liquid interface is stabilized by the nanoparticle-surfactant jamming or change in the micelle morphology. However, the stabilized liquid-liquid structures with the nanoparticle jamming suffer from the lack of internal structures. The recently published work by **Honaryar et al. (2021)** offers a pathway to create internal structures, however, the method requires photopolymerization and structures are not formed spontaneously. In a recent

book published in “*Royal Society of Chemistry, January 2020*” entitled “Bijels: Bicontinuous Particle-stabilized Emulsions”, a comprehensive review on different approaches for creating bicontinuous structures, their utilization in 3D printing systems, and their wide potential applications has been provided. The book concludes by noting the absence of an approach for creating spontaneous and droplet-based 3D printed all-in-liquid materials, i.e., self-derived compartmental emulsification without external energy, similar to what has been reported in the submitted manuscript.

Table S2: Summary of liquid-in-liquid printing publications

Method	Publications	Materials	Advantages	Shortcomings
Nanoparticle-polymer jamming	Toor2017	COOH-Silica/PDMS-NH2	 1. Spontaneous formation of a viscoelastic interfacial layer. 2. Fluid exchange between the printed frame and surrounding media 3. Can be used in many particle-polymer systems 4. Creating multi-layer all-in-liquid fluid channel that enhances the biocompatibility 	 1. Incapability of encapsulating incompatible cargoes (particles) inside the printing frame 2. Lack of internal structure 3. Long-time stability has not been reported
	Lui2017	CNC-OSO3/PS-NH2		
	Shi2018	CNC-OSO3/PS-NH2		
	Forth2018	COOH-Silica, COOH-Au, COOH-CNC /NH2-PDMS-NH2, PDMS-NH2, copolymer		
	Feng2019	Nanoclay/ NH2-PDMS-NH2		
	Toor2019	COOH-Silica/ NH2-PDMS-NH2, PDMS-NH2		
	Qian2020	DNA/ POSS-NH2		
	Sun2020	α -CD-Au/ Azo-PS, Azo-PLLA		
	Wang2022	SPAA/ Ad-PLLA		
	Yin2021	CNC/PDADMAC		
	Kamkar2022	GO/ POSS-NH2		
Lui2022	CNC/POSS-NH2			
Formation of lamella structures	Niroobakhsh2018	CPCI surfactant/oleic acid	Spontaneous formation of a viscoelastic interfacial layer	 1. Incapability of encapsulating incompatible cargoes (particles) inside the printing frame 2. Lack of internal structure 3. Long-time stability can only be achieved by photopolymerization 4. Requires change in micelle morphology
	Niroobakhsh2019	CPCI surfactant/oleic acid		
	Honaryar2021	CPCI- PEGDA, F68-PEGDA/oleic acid		
Emulsion printing in liquid	Zhao2021	Adhesive oil in water emulsions are stabilized with copolymer surfactants functionalized by zwitterionic and ternary amine groups.	 1. Formation of porous internal structures 2. Encapsulation of incompatible materials 	 1. Requires input energy for formation of emulsions 2. It is limited to the gel like emulsions 3. Conventional emulsions suffer from the stability
	Kamkar 2022	Oil in water emulsions are stabilized with GO and are printed an oil phase that contains POSS		
Spongy liquid columns			 1. Formation of porous internal structures 2. Does not require input energy for emulsification 3. Interactions between the printed texture and surrounding media 4. Spontaneous formation of a viscoelastic interfacial layer 5. Encapsulation of incompatible materials 	 2. Requires the presence of surfactant micelles 3. Requires nanoparticles to reduce the interfacial tension to values lower than 0.1 mN/m
			 1. Printed structure remains stable for up to two months 	

Reviewer #2 (Remarks to the Author):

This paper presents a simple and novel approach to create stable liquid filaments of silica nanoparticle dispersions and propose to use them as inks to construct materials consisting a network of droplets. The phenomena is well characterized and demonstrated. The mechanism of forming the liquid filaments with the presence of silica particles is well explained. The strategy of creating liquid columns by in-situ emulsification is general and should be potentially beneficial to any applications that involve highly interconnected structures with high surface-to volume ratios. Therefore, I suggest to accept this manuscript after addressing the following minor comments.

We are pleased to read that the reviewer notes that the phenomena is well characterized and demonstrated. The mechanism of forming the liquid filaments with the presence of silica particles is well explained. The strategy of creating liquid columns by in-situ emulsification is general and should be potentially beneficial to any applications that involve highly interconnected structures with high surface-to volume ratios.” Also, we are happy to read that the reviewer describes the approach as “novel” and recommends that the manuscript be accepted.

Q2.1. Figure 1d is not readable at all. I suggest to decompose it into several ones to make the point clearer.

R2.1. As suggested by the reviewer we added 2D planes of Figure 1d to SI as **Figure S12**.

Figure S12: 2D planes of flow regime morphologies in Figure 1d. (a) Silica concentration-Span concentration, (b) Weber number-silica concentration, and (c) We number-Span concentration.

Q2.2. On page 7, line 84, the author state that “water droplets are Span micelles that are filled with water after being in contact with the aqueous phase”. It would be nice if author can provide any evidence or reference to support this point.

R2.2. The oil phase contains a high concentration of Span 80 surfactants; thus, reverse micelles are formed inside the oil phase. Before the contact with the aqueous phase, the size of the reverse micelle is 3-5 nm (measured with dynamic light scattering). We have clarified our writing to make these parts of our contribution more clear as:

Emulsions are generated by two mechanisms in the presence of silica particles: (i) the presence
of reverse surfactant micelles that intake the aqueous phase and (ii) the low interfacial tension.
Under very low interfacial tension conditions, the interfacial area would spontaneously increase
by forming small droplets close to the interface. In the case of high viscosity oil, the interfacial
area is extended by the formation of droplets and folding, resulting in the penetration of fingers
from one phase into the second phase [40]. Later, the invaded streams turn into small droplets.
The expansion of interfacial area by the penetration of emulsions from the interface to the oil
phase is shown in **Fig. S14** and Supporting Movie 5.

Figure S14: Expansion of the interfacial area and penetration of aqueous phase into the oil phase over time.

High magnification confocal images of the interconnected structures of oil and water that shows the presence of small droplets in the sample. Images are taken from a diluted region. Although there are many small sub-micrometer droplets within the sample, we cannot observe the submicrometric droplets due to the limitation in resolution.

Figure S17: High magnification confocal images of the interconnected structures of oil and water that shows the presence of small droplets in the sample.

Q2.3. On page 10, line 118, the author should first explain the definition and the physic meaning of emulsification time and diffusion time?

R2.3. Following the reviewer’s suggestion, we added the definition of emulsification time and the diffusion time at the beginning of the paragraph and clarify the use of emulsification and diffusion time scales. We refer the reviewer to section 3.3 in the revised manuscript (lines 179-217) and section VII.1 in supporting information for the detailed analysis on filament break-up and discussion on diffusion/emulsification times.

Revised section in the manuscript:

173 **3.3 Universality of in-situ emulsification triggered liquid columns**

174 To unravel the general conditions for the generation of liquid columns, we analyze different
 175 times scales of the process. The residence time, $t_R = L/U$, is defined as the time elapsed from
 176 the injection to the time the injected fluid reaches the bottom of the container, where U is
 177 the injection speed and L is the distance between the nozzle tip and the container base. The
 178 filament breakup can be prevented if the surface-tension-driven Rayleigh-Plateau instability
 is suppressed within the residence time. The time scale of the Rayleigh-Plateau instability
 depends on the viscosity ratio of the inner to the outer fluid and the surface tension of the
 system [47-51]. Decreasing the interfacial tension and increasing the viscosity of the liquid
 phases reduces the disturbance growth rate and slows the development of the instability in the
form of thread breakup and droplet formation.
We calculated the perturbation growth rates and estimated the breakup time for two im-
miscible liquids from the prediction of the Rayleigh-Plateau instability theory [48,50,52] (Sup-
porting information, section VII.1). For the case of clean water (without nanoparticles), the
experiments show droplet formation (i.e., an unstable thread) in all tested Span 80 concen-
trations and injection flow rates (**Fig. S3**). The estimated time from the Rayleigh-Plateau
instability theory is ~ 0.1 s, which has the same order of magnitude as the droplet formation
(**Fig. S3**). Hence, the experiments support the theory for clean water-Span solutions. Adding
nanoparticles to water considerably reduces the effective interfacial tension, which consequently
increases the estimated time for the droplet breakup to ~ 1 s for the two higher silica particle
concentrations (2.0 and 4.0 wt.%). So, for silica dispersion-Span solution systems, one expects
that the droplet formation occurs for a residence time greater than one second. However, ex-
periments show that the silica dispersion thread remains stable without droplet formation in
silica 2.0 wt.%-Span 40.0 wt.% and silica 4.0 wt.%-Span 10.0-40 wt.% (**Figs. S5-6**) for times
much greater than 1 s. Therefore, other mechanisms, besides the low interfacial tension of silica
dispersion-span micellar solution and the high viscosity of the outer fluid, could contribute to
the attenuation of Rayleigh-Plateau instability within the time frame of experiments.
The rapid in-situ formation of emulsions at silica dispersion-Span 80 micellar solution in-
terfaces, as described in **Fig. 2**, may suppress the Rayleigh-Plateau instability in our systems.
As revealed by experiments (**Figs. S5-6**), liquid columns are formed at an intermediate range
of flow rates (residence time). For shorter time scales, emulsions do not have enough time to be
generated at the interface, thus, for such small residence times, the emulsion layer has not been
formed fully to cover the oil-water interface and consequently cannot prevent the Rayleigh-
Plateau instability. We refer to the smallest residence time in which liquid columns can be
generated as the emulsification time (t_E). Despite the fact that emulsions have enough time for
their formation at the longest residence times, we cannot see the formation of stabilized liquid
filaments at these residence times either. Thus, aside from emulsification, another mechanism,
i.e., emulsion drop removal from the interface, is involved, which affects the stabilization of
liquid filaments. Since we do not have any external source of flow, the main mechanism for
the detachment of emulsions from the interface can be the diffusion of swollen reverse micelles
(emulsion droplets) from the interface into the bulk. The longest residence time in which the
liquid columns are formed is considered as diffusion time (t_D). In summary, in our system, the
Rayleigh-Plateau instability is attenuated by the in-situ formation of an emulsion layer at the
aqueous phase-oil interface.

Added section in supporting information:

VII.1 Rayleigh-Plateau instability analysis

We calculate the break-up time of the liquid filament in the presence and absence of nanoparticles with the assumption that no interfacial materials are formed at the water-oil interface. So, only the interfacial tension and fluids' viscosities are the contributing factor in the filament instability. For a liquid filament (radius of R , viscosity of μ_i) flowing through a fluid with the

viscosity of μ_e in a container with the width of W , the rate of disturbance grow is [7,8]:

$$\omega = \left(\frac{\gamma}{16\mu_e W} \right) \left[\frac{F(x, \lambda)(k^2 - k^4)}{x^9(1 - \lambda^{-1}) - x^5} \right] \quad (\text{Eq. S.1})$$

k is the dimensionless wavenumber of the perturbation and it is considered as 0.257 based on the viscosity ratio $\mu_i/\mu_e = O(0.001)$ [9], x is the dimensionless radius of the thread ($x = R/W$), λ is the viscosity ratio μ_i/μ_e , and $F(x, \lambda) = x^4(4 - \lambda^{-1} + 4\ln x) + x^6(-8 + 4\lambda^{-1}) + x^8(4 - 3\lambda^{-1} - (4 - 4\lambda^{-1})\ln x)$ [7,8]. In the absence of nanoparticles (DI water), the equation predicts a maximum growth rate of $w(k = 0.257) = 61 \text{ s}^{-1}$. Assuming that perturbations at the entrance initially are of nanometer size and perturb the radius as $r = r_0 + \epsilon_0 e^{i(k/r_0)z + wt}$ until $r_0 = \epsilon_0 e^{wt}$, the break-up time is predicted as $t = \omega^{-1} \ln(200 \times 10^3) = 0.2 \text{ s}$. In the presence of silica particles (in systems that the liquid filament is generated as an example Span 20.0 wt. %-Silica 4.0 wt. %), $w(k = 0.257) = 2.5 \text{ s}^{-1}$, the predicted break-up time is $t = 4.9 \text{ s}$. Unlike the systems without nanoparticles, the calculated instability time in the presence of nanoparticles does not match the flow regime transition in **Fig. S6**. Thus, we concluded the low interfacial tension of the

Reviewer #3 (Remarks to the Author):

The submitted manuscript reports on liquid-in-liquid printing of filaments that are stabilized by emulsions that form when an aqueous solution containing silica nanoparticles come in contact with an oil phase containing Span 80 micelles. Four different flow regime morphologies are reported and include single droplets, bead-on-a-string, column, and connected. The transition from single droplets to the three other morphologies is a result of the addition of silica nanoparticles and increased injection speed. Optical and scanning electron microscopy images indicate that the emulsion morphologies appear different between DI water and the aqueous solution containing silica nanoparticles. The formation of smooth liquid columns is hypothesized to be a result the residence time is between emulsification and diffusion times, and is controlled by the injection speed. A few examples are shown indicating that it is possible to print liquid filaments.

Overall, the liquid-in-liquid printing results are interesting and reveal a new method for printing liquid structures. Although some of the stated claims are supported by the results, there are areas that need to be addressed and revised. The major concern is the classification that the microemulsion morphology is bicontinuous when silica nanoparticles are present. Once the authors address the minor comments below, the manuscript is publishable.

We are pleased to read that the reviewer wrote “the liquid-in-liquid printing results are interesting and reveal a new method for printing liquid structures” and recommends publication once minor comments are clarified.

Comments

Q3.1. How are the authors defining the bicontinuous morphology? In surfactant and diblock copolymer literature, the bicontinuous microemulsion structure is defined as having zero mean curvature and negative Gaussian curvature. The microemulsion phase in the SEM images look to have non-zero mean curvature. Therefore, it is highly unlikely that a bicontinuous morphology forms.

R3.1. We consider the bicontinuous phase as an interconnected zone of oil and emulsion phase, depicted in **Figures S15 and S20**, not the emulsion zone by itself. In these images, the interfacial curvature between the oil and emulsion phase is close to zero (they form parallel zones), and similar structures have been reported for bijels [**Huage et al. (2017)**]. For clarification purposes, we have more details about “the bicontinuous phase” in the revised version as:

has a different structure than the one formed from DI water (**Fig. 2a. 8'**). In addition to
the generated micro-scale droplets, we notice the coexistence of interconnected zones of oil and
~~microemulsion~~ emulsion-rich phases, naming as a bicontinuous in this study, throughout the
sample. In fact, the silica nanoparticle dispersion generates a bicontinuous phase that can be a

Figure R1 a) interconnected structures of oil and emulsion phase. b) binarized image that clearly shows the oil phase is not in the forms of spherical droplets and it has arbitrary structures.

Q3.2. An alternative claim is that the silica nanoparticles preferentially organize that the aqueous/oil interface, creating a more elastic droplet interface that jams the system. Nanoparticles are known to be effective emulsifying agents. It is hard to determine from the SEM images, but it looks like the droplet diameter decreases with added silica nanoparticles. This would help confirm the emulsifying characteristics of the nanoparticles.

R3.2. We thank the reviewer for their attention to details and the concise comment about reducing the droplet size upon adding the particles. We measured the emulsion droplet size from Cryo-SEM images and added the figure to supporting document as **Figure S16**. We also added a comment in the manuscript in this regard as:

~~Fig. 1b~~ are not formed due to a change in reverse micelle morphology. The size of the emulsion
 droplets reduces upon adding silica particles and increasing the concentration from $\sim 5 \mu\text{m}$
 for DI water to $\sim 1 \mu\text{m}$ for 4.0 wt.% silica dispersion (**Fig. S16**), indicating the enhanced
 emulsification properties of the system. Inspection of the silica-Span micellar solution interface

Figure S16: Emulsion droplet size (droplet diameter) over silica concentration. The droplet size is obtained from Cryo-SEM microscopic images.

Q3.3. The rapid decrease in interfacial tension with the addition of nanoparticles suggests that they reside at the aqueous/oil interface. How does the stabilization time of the interfacial tension change of with silica nanoparticle wt.%?

R3.3. We added the dynamic interfacial tension data for all the tested concentrations in **Figure S2**. In the presence of particles, the interfacial tension values are smaller and the reduction in interfacial tension is faster and reach the equilibrium state in less than a few minutes. The stabilization time is a decreasing function of silica nanoparticle concentration.

Figure S2: Dynamic interfacial tension. (a) Span 5.0 wt.%, (b) Span 10.0 wt.%, and (c) Span 20.0 wt.%.

Q3.4. Page 7, Line 90: There may not be a change in the Span micelle morphology, but the optical and scanning electron microscopy images do not probe nanometer length scales that would be necessary to support the claim. Please revise. The comment is also made in the conclusion section.

R3.4. We agree with the reviewer's comment that we may have lost some details in the change of the morphology on a smaller scale. Thus, as suggested, we remove this claim.

Q3.5. There is a recent article showing that it is possible to print robust liquid filaments with internal phase nanostructures using both ionic and non-ionic surfactants (Macromol. Rapid Commun. 2021, 42, 2100445). Thus, the statement in the conclusion that 3D printing of micellar solutions is limited to ionic surfactants is not correct. The authors seemed to have missed the reference.

R3.5. Thank you for pointing out this reference. We discussed this paper in the revised manuscript and adjusted the related sections accordingly as:

Structured liquids with submicrometer domains can be achieved by the solvent transfer
induced phase separation (STRIPS) approach [28,29]. A homogeneous mixture of oil-water-
alcohol-surfactant-particle is rapidly injected into the continuous water phase. The phase sep-
aration starts upon the injection, where the alcohol is extracted into a continuous phase from
the mixture. The presence of particles results in the arrest of the mixture in a transition zone
of bicontinuous structures during the phase separation [29]. Recently, internal structures inside
the all-in-liquid printed materials are accomplished with photocurable polymers [30]. Oil-water
interfaces are stabilized by the interactions of surfactant assemblies, i.e. micelles, with fatty
acids, where micelle morphology is changed from spherical to lamella and a gel phase is formed
at the interface [9,30]. In the presence of photocurable polymers inside the internal phase,
submicrometer regions are formed after photopolymerization [30]. STRIPS approach is limited
to the use of a ternary mixture and photopolymerization eliminates the fluidity of the printed
material. Besides, these approaches do not form an interconnected network of droplets similar
to those with emulsions. Thus, an approach for creating spontaneous and droplet-based 3D
printed all-in-liquid materials, i.e., self-derived compartmental emulsification is required.

Reviewer #4 (Remarks to the Author):

In this manuscript, the authors study the morphological states and flow instabilities of ternary/quaternary mixtures of mineral oil-Span-water (w or w/o Silica nanoparticles) with some applications in printing. There are some concerns regarding the validity of the proposed mechanism and the significance of the work. Also, the connection between fluid dynamic results and direct applicability of them to printing is not discussed. I have following specific comments.

Major comments:

Q4.1. The discussion about underlying stabilization phenomena is not convincing and the arguments are not completely correct or well supported. For example, the term microemulsion are misused (e.g., figure 2 and line 166, line 83). In the colloids field, the microemulsion is referred to a thermodynamically-equilibrated phase that possess features in the order of only a few nanometers (such as micellar, inverse micellar) thus not possible to be captured in the micro-meter scale as in figures 2, S12, and S19. There are major differences in terms of stability and scales in emulsion and microemulsion definitions that are not properly addressed throughout the paper. Furthermore, it is claimed that the in-situ emulsification could be the underlying mechanism for stabilization of interface, however emulsified phases are often unstable with insufficient mechanical properties (such as water-in-oil droplets or oil-in-water emulsions), unless they have some internal nanostructures, that are not addressed here. It is also relevant to discuss the possible phase changes in emulsification or formation ouzu effects that are commonly used in the well-established area of nanoprecipitation. Also, it is not clear how the authors concluded that the micelle morphology remains unchanged in the presence of absence of nanoparticles (line 90) by looking on the SEM pictures (which has droplets of micrometer sizes), since the micelles (usually in the order of a few nanometers) cannot be captured in that image.

R4.1. We thank reviewer 4 for the comprehensive and constructive comments that helped us to clarify the manuscript, including small issues of language in the literature that are noted above. We addressed each concern raised by the reviewer as follow:

i) For example, the term microemulsion are misused (e.g., figure 2 and line 166, line 83). In the colloids field, the microemulsion is referred to a thermodynamically-equilibrated phase that possess features in the order of only a few nanometers (such as micellar, inverse micellar) thus not possible to be captured in the micro-meter scale as in figures 2, S12, and S19. There are major differences in terms of stability and scales in emulsion and microemulsion definitions that are not properly addressed throughout the paper.

Microemulsions are generally referred to as systems where the dispersed domain size is less than 10 nm; thus, they are transparent [Lopez et al. (2002)]. We understand that, although in our case emulsions are formed spontaneously, they are not microemulsions. We changed the microemulsion terminology to emulsion throughout the manuscript (though the size scale of the emulsions we observe are micrometers), as suggested by the reviewer.

ii) Furthermore, it is claimed that the in-situ emulsification could be the underlying mechanism for stabilization of interface, however emulsified phases are often unstable with insufficient mechanical properties (such as water-in-oil droplets or oil-in-water emulsions), unless they have some internal nanostructures, that are not addressed here.

Mixing two immiscible phases with a shaker/homogenizer/sonicator generates a kinematically stable dispersion that undergoes destabilization over time. However, in our system, emulsions are

generated spontaneously at the interface, and they are thermodynamically stable as they are formed without input energy. Following this comment, we added a discussion on emulsification mechanisms in our systems as:

Emulsions are generated by two mechanisms in the presence of silica particles: (i) the presence
of reverse surfactant micelles that intake the aqueous phase and (ii) the low interfacial tension.
Under very low interfacial tension conditions, the interfacial area would spontaneously increase
by forming small droplets close to the interface. In the case of high viscosity oil, the interfacial
area is extended by the formation of droplets and folding, resulting in the penetration of fingers
from one phase into the second phase [40]. Later, the invaded streams turn into small droplets.
The expansion of interfacial area by the penetration of emulsions from the interface to the oil
phase is shown in **Fig. S14** and Supporting Movie 5.

Figure S14: Expansion of the interfacial area and penetration of aqueous phase into the oil phase over time.

A concentrated layer of emulsion at the interface is sufficient to form the liquid filament as the liquid filaments remain intact under injection/re-injection cycles, as shown in **Figure 4**.

iii) It is also relevant to discuss the possible phase changes in emulsification or formation ouzo effects that are commonly used in the well-established area of nanoprecipitation.

We added a discussion on the formation of bicontinuous emulsions with phase change. For the ouzo effect, the presence of a third phase with mutual solubility with two other liquid phases is required. For example, the emulsification by the ouzo effect can occur when a mixture of oil and ethanol is placed in contact with water. As alcohol diffuses from the oil into the water, it carries some oil into the water; by further diffusion of alcohol, the oil phase in water forms small droplets. In our case, we do not have a component with mutual solubility; thus, we do not have the ouzo effect.

The formation of bicontinuous structures in particle-laden systems has been reported
through spinodal decomposition [40] or emulsion phase inversion [41–43]. In spinodal de-
composition, a single-phase liquid undergoes de-mixing where a multiphase mixture is gen-
erated. Particles segregate to the interfaces and stabilize the mixture in an out-of-equilibrium
state [40,44]. Interconnected structures can also be generated through temperature [45] or con-
centration [41–43] driven phase emulsion phase change. The hydrophilicity of thermo-responsive
particle is decreased upon increasing the temperature, hence, the initially particle-stabilized oil
in water (O/W) emulsions turn into water in oil (W/O) emulsions. Bicontinuous or multiple
emulsions are formed at temperatures close to the inversion point [45]. The phase inversion can
also occur for partially hydrophobic silica nanoparticles that are initially dissolved in the oil
through phase inversion at high particle concentrations (above 1.0 wt.%) [41–43]. Lodux HS
40 silica particles are not thermo-responsive and particle concentration is constant in each test.
Thus, the generated bicontinuous zones in our experiments differ from reported structures with
particles throughout the literature.

iv) Also, it is not clear how the authors concluded that the micelle morphology remains unchanged in the presence of absence of nanoparticles (line 90) by looking on the SEM pictures (which has droplets of micrometer sizes), since the micelles (usually in the order of a few nanometers) cannot be captured in that image.

We drew this conclusion from the SEM images. To clarify, we meant that there was not any transition to lamella structures. However, as there might be some changes in morphology at the

nm scale, which we cannot resolve, we edited the statement and removed the comment about the micelle morphology.

Q4.2. It is mentioned that the interfacial layer has elastic/viscoelastic properties in the paper. However, no characterization on interfacial properties has been performed to back it up. The authors are advised to perform necessary characterization such as interfacial rheology at the liquid-liquid interface or bulk rheology on isolated gel-like samples. Furthermore, the terms “elastic” and “viscoelastic” are used interchangeably (in the main text and in the supporting information). These two terms are not clearly the same. Use of interfacial and bulk rheology is needed to determine whether mechanical properties of interfacial materials suggest an elastic or viscoelastic behavior.

R4.2. Following the reviewer's comment, we performed rheological characterization to determine the mechanical properties of the interfacial layer. We used an Anton Paar-DHR-2 Rheometer using cone and plate with a cone diameter of 20 mm and angle of 2 deg. First, the aqueous and oil phases were placed in contact for one week to generate enough samples for the measurement. Then, the sample was collected from the interface, and the rheology experiment was conducted for the generated bulk material at the interface and results are presented in **Figure S19**. According to the measurements, the samples show both viscous and elastic properties; thus, they are viscoelastic, and we use this terminology throughout the manuscript.

V Rheological properties of emulsions

We collected samples from the interface of silica 2.0 and 4.0 wt.% after the contact with various Span concentrations (1.0 to 40.0 wt.%), and conducted oscillatory shear experiments, i.e., frequency sweep test at the fixed strain of 0.2 % and presented the results in **Fig. S19**. For the samples collected from the interface of 2.0 wt.% silica dispersion, both elastic and viscous moduli increase by increasing the Span concentration and the angular frequencies. The sample generated from silica 2.0 wt.%-Span 40.0%, which can form a stable liquid filament, has the largest elastic and viscous moduli at the highest frequencies (**Fig. S19a**). Generated samples at silica 2.0 wt.% for all tested Span concentration interfaces, silica 4.0 wt.%-Span 1.0 wt.%, and silica 4.0 wt.%-Span 5.0 wt.%, have a cross-over point where at low frequencies the elastic modulus is higher than the viscous modulus. The generated samples at silica 4.0 wt.%-Span 10.0, 20.0, and 40.0 wt.% interfaces have an elastic modulus greater than 100 Pa, which is always greater than or equal to the viscous modulus. Also, the elastic modulus for these cases reaches a plateau at the low frequencies, an indicator of gel-like materials [4]. For the two highest concentrations (silica 4.0 wt.%-Span 20.0 and 40.0 wt.%), the elastic and viscous moduli have close values indicating that the materials have a gel-like behaviour. Similar viscoelastic responses have been reported for the Bijles formed from jamming of nanoparticle-surfactants at the oil-water interfaces [5].

Figure S19: Rheological properties of the generated interfacial material at the aqueous phase-Span micellar solution interfaces. (a) Silica 2.0 wt.% and (b) silica 4.0 wt.%. The filled and empty symbols represent the elastic and viscous moduli, respectively.

Q4.3. The manuscript shows interesting observations and exhaustive fluid dynamic data for a system of oil-surfactant-water (w or w/o nanoparticles). However, the advantages of the current system compared to other relevant systems are not well supported. The authors are advised to include convincing arguments concerning the novelty of their work. In doing so, the introduction should be more comprehensive covering current approaches in liquid-in-liquid printing (e.g., [1-2,4]) and all various underlying mechanisms as well as a discussion section covering comparisons of their systems with others. The author mentioned a few comparisons somewhere else in the paper (jamming, phase transition, section 3.1) or SI (emulsion-based in table s2), but not discussed it thoroughly in the introduction and the discussion. Also, some of the advantages that claimed about the system are not convincing. For example, in table S2 (and throughout the main text) it is claimed that the system is capable of encapsulating cargos and method 1 and 2 are not. However, they have not performed any encapsulation test to support that. Also, the stability that claimed as the strength of their work (in table B2 and main text line 169) are not supported or at least not illustrated in Figure S18. Structures seem they have lost their shape (such as letter F, L). Another claim about the internal structure and porosity of the structure has 2 not been tested (e.g., for a good porous example:[3]). It is not clear what length scale they are talking about for the internal structures. For example, they mentioned that the internal structure is not available for the method 1 in Table S2, however, the recent publication from that group supports the formation of internal nanostructure:

1. Forth, J., Liu, X., Hasnain, J., Toor, A., Miszta, K., Shi, S., ... & Russell, T. P. (2018). Reconfigurable printed liquids. *Advanced Materials*, 30(16), 1707603.
2. Lin, D., Liu, T., Yuan, Q., Yang, H., Ma, H., Shi, S., ... & Russell, T. P. (2020). Stabilizing Aqueous Three-Dimensional Printed Constructs Using Chitosan-Cellulose Nanocrystal Assemblies. *ACS applied materials & interfaces*, 12(49), 55426-55433.

3. Sears, N. A., Wilems, T. S., Gold, K. A., Lan, Z., Cereceres, S. N., Dhavalikar, P. S., ... & Cosgriff-Hernandez, E. M. (2019). Hydrocolloid inks for 3D printing of porous hydrogels. *Advanced Materials Technologies*, 4(2), 1800343.

4. Honaryar, H., LaNasa, J. A., Lloyd, E. C., Hickey, R. J., & Niroobakhsh, Z. (2021). Fabricating Robust Constructs with Internal Phase Nanostructures via Liquid-in-Liquid 3D Printing. *Macromolecular Rapid Communications*, 42(22), 2100445.

R4.3. Thank you for the attentive comment. We broke down the comment and provided responses to each question separately as follow:

i) Introduction

We revised the introduction and added a detailed literature review on all-in-liquid printing systems and highlighted the novelty of our work.

~~Recently developed~~ Liquid-in-liquid printed materials [6,10-14] have many potential applica-
tions in energy storage [4,5], microreactors [7], and for creating biomimetic materials [8,13].
These types of materials are generated ~~by the jamming of nanoparticles at the oil-water interface~~
by application of an electrical field [16], using molding [11,17], or direct ink writing (DIW) print-
ing techniques [10,12]. Liquid-fluid interfaces can be arrested in desired non-equilibrium shapes
by the adsorption of colloidal particles. For instance, bicontinuous structures, called bijels, are
formed from the spinodal decomposition of a binary liquid mixture containing amphiphilic par-
ticles [18,19]. Spherical bubbles covered with particles [20] and liquid droplets coated with the
assembly of nanoparticles and end-functionalized polymers [16] can be redesigned into various
anisotropic shapes by the particles' jamming at the interface. The driving force for reducing the
surface area jams nanoparticles at the interface and creates a viscoelastic layer that locks the
system in any arbitrary structure. Nanoparticle jamming at the oil-water interface can prevent
jet break-up [10], where stabilized liquid filaments are used to create reconfigurable all-in-liquid
composite materials of oil-water [6,11,12,21–23] and water-water [13,24]. Particle jamming
technique is applicable for a wide range of liquid viscosities and many particle-polymer combi-
nations, and enable the exchange of materials between the printed texture and the surrounding
fluid. However, the generated materials with particle jamming approach lack multiscale poros-
ity created by emulsion-based inks used in 3D printed solid structures [25–27].
Structured liquids with submicrometer domains can be achieved by the solvent transfer
induced phase separation (STRIPS) approach [28,29]. A homogeneous mixture of oil-water-
alcohol-surfactant-particle is rapidly injected into the continuous water phase. The phase sep-
aration starts upon the injection, where the alcohol is extracted into a continuous phase from
the mixture. The presence of particles results in the arrest of the mixture in a transition zone
of bicontinuous structures during the phase separation [29]. Recently, internal structures inside
the all-in-liquid printed materials are accomplished with photocurable polymers [30]. Oil-water
interfaces are stabilized by the interactions of surfactant assemblies, i.e. micelles, with fatty
acids, where micelle morphology is changed from spherical to lamella and a gel phase is formed
at the interface [9,30]. In the presence of photocurable polymers inside the internal phase,
submicrometer regions are formed after photopolymerization [30]. STRIPS approach is limited
to the use of a ternary mixture and photopolymerization eliminates the fluidity of the printed
material. Besides, these approaches do not form an interconnected network of droplets similar
to those with emulsions. Thus, an approach for creating spontaneous and droplet-based 3D
printed all-in-liquid materials, i.e., self-derived compartmental emulsification is required.

ii) For example, in table S2 (and throughout the main text) it is claimed that the system is capable of encapsulating cargos and method 1 and 2 are not. However, they have not performed any encapsulation test to support that.

We made this claim as we have reverse micelles in the system, and they are capable of encapsulating aqueous phases and hydrophilic components. We added high-resolution confocal microscopy images in **Figure S17** showing the nanoparticle dispersion inside the emulsion droplets. This shows the encapsulation capability as a hydrophilic phase (silica dispersion) is encapsulated inside a hydrophobic phase (oil). In the image, the green color shows the oil phase, and the red color shows the nanoparticle dispersion tagged with Rod-amine B dye.

Figure S17: High magnification confocal images of the interconnected structures of oil and water that shows the presence of small droplets in the sample.

iii) Another claim about the internal structure and porosity of the structure has 2 not been tested (e.g., for a good porous example:[3]).

We analyzed the porosity of the 3D printed structures in **Figures S25-28**. The porosity increases sharply within first 20 minutes and it reaches a plateau of ~60%.

iv) Stability

Without polymerization, previous works reported stability in terms of a few hours. Our printed liquid letters remain stable up to two months. Although, they lost some of their sharp edges over time, they have not dissolved completely. The stability of the printed letters is quantified in **Figure 4b**.

v) It is not clear what length scale they are talking about for the internal structures. For example, they mentioned that the internal structure is not available for the method 1 in Table S2, however, the recent publication from that group supports the formation of internal nanostructure:

We have submicrometric domain internal structures made of emulsion droplets as illustrated in SEM images and their quantification (**Figure 2** and **Figures S15-16**). We discussed the publications on the internal structure, suggested by the reviewer, in the introduction.

Q4.4. The description of initial oil phases and their interpretation are not clear. For example, 20 wt% span solution (in mineral oil) is claimed to be a micellar solution, however, since the continuous phase is an oil, one can expect the formation of reversed micelles and not micelles. Also, the rheological properties of that phase suggested a shear thinning behavior (Fig S1b), however, it is unlikely that the micellar phases alone can show non-Newtonian behavior. If possible, proper characterization (such as x-ray scattering) should be used to support the internal structures of such phase. Authors need to integrate the structure and properties into understanding the underlying phenomena in the main text discussion.

R4.4. In this manuscript, by micelle, we meant reverse micelles as the micelles are inside the oil phase. To avoid confusion, we changed micelle to reverse micelle throughout the manuscript. To address the comment regarding the micellar (reverse micellar) solution's shear-thinning behaviour, we measured the mineral oil's viscosity (without any surfactant in the system). The rheology plot is similar to the one with surfactant micelles. We checked the original papers on the shear viscosity of the mineral oil. Within the range of measurements (3.3-20 1/s) [**Jaona (2011)**], the plot shows a Newtonian behaviour as is the case for our measurement. However, the viscosity has not been reported at a lower shear rate. Thus, we believe the non-Newtonian behaviour of the micellar solutions at shear rates lower than 1 (1/s) is related to the properties of the mineral oil, and it is not the effect of micelles.

We cannot capture the shape of the micelles in the absence of water as they are around 5 nm as confirmed through dynamic light scattering measurements. However, the emulsion drops in the oil phase are spherical. To characterize the structure of the reversed micelles using x-ray scattering in a liquid media, we need a SAXS instrument with the pinhole of 12IDB. Unfortunately, we currently do not have access to this instrument.

Figure R2: Shear viscosity of the mineral oil sample (without any surfactant).

Q4.5. The title should be more descriptive of what is presented in the paper, i.e., include the underlying mechanisms or process. Also, the title includes “printing” which is not investigated at all, as no 3D model was used for printing, rather all the liquid structures were created. Furthermore, it is not clear why the terms "spongy", "all-in-liquid" are used, for instance, spongy texture is not well illustrated in Fig 4f. The spongy structure actually is a well-defined nanostructure in colloids field and use of spongy liquid here could be misleading.

R4.5. Thank you for this suggestion. We changed the title to “Spongy all-in-liquid materials: Attenuating *Rayleigh-Plateau* instability by in-situ formation of emulsions at oil-water interfaces” to better describe the novelty of the work. Spongy is used to represent the texture of interfacial layer which is made of emulsions. Furthermore, the emulsification can proceed and the entire printed texture can turn to emulsion. We investigated the printing in **Figure 4** for liquid letters and liquid-fluidic channel. The spongy texture is quantified by the term of porosity in **Figure S28**.

Q4.6. The authors assigned emulsification time t_e and diffusion Damkoeler time t_D based on some initial assumption on flow column morphology (as in Section 3.3 and fig 3a), which is questionable. Authors claimed that the time scale t_R should be between emulsification time t_e and diffusion Damkoeler time t_D as the criterion for the formation of the liquid column. The argument to support that criterion is not clear. Also, there is almost no connection between the flow column morphological states observed in section (3.1) with printing application in section (3.3). It is advised to discuss what flow morphologies can be used for printing and what conditions should be considered.

R4.6. Flow morphologies in **Figure 1** and **Figures S3-6** are identified based on the Fourier transfer analysis as described in supporting document **section III**. The Fourier analysis forms the basis for the quantified criteria in distinguishing the flow patterns as opposed to visual examination of

images. Thus, the time scales, namely t_D and t_E are found experimentally and based on Fourier analysis to identify the transition from BOAS to column and column to connected flow regimes, respectively. Inspection of these time scales and their relations to the fluid properties reveals that t_E and t_D are directly proportional to the equilibrium interfacial tension (γ_{eq}) and the micellar solution viscosity (μ), respectively. Through dimensional analysis and with the assumption that the equilibrium interfacial tension and the micellar solution viscosity are the dominant factors in emulsification and diffusion time scales, we could develop two correlations describing the time scales as $t_D = (6\pi a \ell^2 / k_B T) \mu$ and $t_E = (a^2 / \kappa k_B T) \gamma_{eq}$; where a , ℓ , k_B , and T are, respectively, the micelle diameter, diffusion length scale, Boltzmann constant, and temperature (see supporting information **section VII.1-2** and the main text **section 3.3**). We used these correlations to develop the generalized map describing the flow morphologies and the desired column regime as depicted in **Figures 3c-d**. Following this comment, we added a section in the manuscript on the related mechanisms of filament stability and clarify the emulsification and diffusion time scales, and also provided a discussion on the effect of the third phase (emulsion) formation at the oil-water interface on the attenuation or promotion of *Rayleigh-Plateau* instability. The added discussion is: Revised section in the manuscript:

**3.3 Universality of in-situ emulsification triggered liquid columns**
To unravel the general conditions for the generation of liquid columns, we analyze different
175 times scales of the process. The residence time, $t_R = L/U$, is defined as the time elapsed from
176 the injection to the time the injected fluid reaches the bottom of the container, where U is
177 the injection speed and L is the distance between the nozzle tip and the container base. The
178 filament breakup can be prevented if the surface-tension-driven Rayleigh-Plateau instability
is suppressed within the residence time. The time scale of the Rayleigh-Plateau instability
depends on the viscosity ratio of the inner to the outer fluid and the surface tension of the
system [47–51]. Decreasing the interfacial tension and increasing the viscosity of the liquid
phases reduces the disturbance growth rate and slows the development of the instability in the
form of thread breakup and droplet formation.
We calculated the perturbation growth rates and estimated the breakup time for two im-
miscible liquids from the prediction of the Rayleigh-Plateau instability theory [48,50,52] (Sup-
porting information, section VII.1). For the case of clean water (without nanoparticles), the
experiments show droplet formation (i.e., an unstable thread) in all tested Span 80 concen-
trations and injection flow rates (**Fig. S3**). The estimated time from the Rayleigh-Plateau
instability theory is ~ 0.1 s, which has the same order of magnitude as the droplet formation
(**Fig. S3**). Hence, the experiments support the theory for clean water-Span solutions. Adding
nanoparticles to water considerably reduces the effective interfacial tension, which consequently
increases the estimated time for the droplet breakup to ~ 1 s for the two higher silica particle
concentrations (2.0 and 4.0 wt.%). So, for silica dispersion-Span solution systems, one expects
that the droplet formation occurs for a residence time greater than one second. However, ex-
periments show that the silica dispersion thread remains stable without droplet formation in
silica 2.0 wt.%-Span 40.0 wt.% and silica 4.0 wt.%-Span 10.0-40 wt.% (**Figs. S5-6**) for times
much greater than 1 s. Therefore, other mechanisms, besides the low interfacial tension of silica
dispersion-span micellar solution and the high viscosity of the outer fluid, could contribute to
the attenuation of Rayleigh-Plateau instability within the time frame of experiments.
The rapid in-situ formation of emulsions at silica dispersion-Span 80 micellar solution in-
terfaces, as described in **Fig. 2**, may suppress the Rayleigh-Plateau instability in our systems.
As revealed by experiments (**Figs. S5-6**), liquid columns are formed at an intermediate range
of flow rates (residence time). For shorter time scales, emulsions do not have enough time to be
generated at the interface, thus, for such small residence times, the emulsion layer has not been
formed fully to cover the oil-water interface and consequently cannot prevent the Rayleigh-
Plateau instability. We refer to the smallest residence time in which liquid columns can be
generated as the emulsification time (t_E). Despite the fact that emulsions have enough time for
their formation at the longest residence times, we cannot see the formation of stabilized liquid
filaments at these residence times either. Thus, aside from emulsification, another mechanism,
i.e., emulsion drop removal from the interface, is involved, which affects the stabilization of
liquid filaments. Since we do not have any external source of flow, the main mechanism for
the detachment of emulsions from the interface can be the diffusion of swollen reverse micelles
(emulsion droplets) from the interface into the bulk. The longest residence time in which the
liquid columns are formed is considered as diffusion time (t_D). In summary, in our system, the
Rayleigh-Plateau instability is attenuated by the in-situ formation of an emulsion layer at the
aqueous phase-oil interface.

Added section in supporting information:

VII.1 Rayleigh-Plateau instability analysis

We calculate the break-up time of the liquid filament in the presence and absence of nanoparticles with the assumption that no interfacial materials are formed at the water-oil interface. So, only the interfacial tension and fluids' viscosities are the contributing factor in the filament instability. For a liquid filament (radius of R , viscosity of μ_i) flowing through a fluid with the

viscosity of μ_e in a container with the width of W , the rate of disturbance grow is [7,8]:

$$\omega = \left(\frac{\gamma}{16\mu_e W} \right) \left[\frac{F(x, \lambda)(k^2 - k^4)}{x^9(1 - \lambda^{-1}) - x^5} \right] \quad (\text{Eq. S.1})$$

k is the dimensionless wavenumber of the perturbation and it is considered as 0.257 based on the viscosity ratio $\mu_i/\mu_e = O(0.001)$ [9], x is the dimensionless radius of the thread ($x = R/W$), λ is the viscosity ratio μ_i/μ_e , and $F(x, \lambda) = x^4(4 - \lambda^{-1} + 4\ln x) + x^6(-8 + 4\lambda^{-1}) + x^8(4 - 3\lambda^{-1} - (4 - 4\lambda^{-1})\ln x)$ [7,8]. In the absence of nanoparticles (DI water), the equation predicts a maximum growth rate of $w(k = 0.257) = 61 \text{ s}^{-1}$. Assuming that perturbations at the entrance initially are of nanometer size and perturb the radius as $r = r_0 + \epsilon_0 e^{i(k/r_0)z + wt}$ until $r_0 = \epsilon_0 e^{wt}$, the break-up time is predicted as $t = \omega^{-1} \ln(200 \times 10^3) = 0.2 \text{ s}$. In the presence of silica particles (in systems that the liquid filament is generated as an example Span 20.0 wt. %-Silica 4.0 wt. %), $w(k = 0.257) = 2.5 \text{ s}^{-1}$, the predicted break-up time is $t = 4.9 \text{ s}$. Unlike the systems without nanoparticles, the calculated instability time in the presence of nanoparticles does not match the flow regime transition in **Fig. S6**. Thus, we concluded the low interfacial tension of the

For printing we need to have stable liquid filaments, the criteria for the formation of stable liquid filaments are defined in the phase diagram in Figures 3c-d.

Minor comments:

Q4.1. In line 29, the authors reported the average speed for injection, but there is no mention of how they calculate those values.

R4.1. We added the calculation of the average speed (it is the injection flow rate divided by the cross-sectional area of the needle).

Q4.2. In line 138 specifically and throughout the manuscript in general, authors report that at an interfacial tension higher than 0.6 mN/m, the emulsification time is 0.2-0.5 s. From what we can see based on Figure 3b, we believe they meant 0.06, not 0.6 mN/m.

R4.2. Thank you, we fixed it.

Q4.3. In the “methods and material” section and for the “viscosity measurements”, no information is provided about the rheometer geometry that was used (parallel plate or cone & plate). Also, it is advised to include more details about the test procedure (what shear rate range was used for setting up the test, what gap was used).

R4.3. We added the details of the measurements as:

³⁶⁷ **5.7 Viscosity measurements**

³⁶⁸ The viscosities of the aqueous phases (silica dispersions), micellar solutions, and **microemulsions**
³⁶⁹ **emulsion** (bicontinuous phase) are measured at shear rates of $10^{-3} - 10^3$ (1/s) using Discov-
³⁷⁰ ery Hybrid Rheometer (DHR-2, Anton Paar). The rheological properties of the samples are
³⁷¹ measured in a cone and plate geometry with a cone diameter of 20 mm and angle of 2 degree.

Q4.4. In the supporting information, the shear rate is calculated in the experiments using “ $8U/di$ ”. Where is this equation is coming from and can authors provide a reference for it?

R4.4. It is the shear rate for the Newtonian fluid flowing through a pipe. Equation 3.16 in [2017Draby]. We added the reference in the text.

Q4.5. Many pictures (especially SEM pictures in main text and SI, e.g., fig1b7’) are too small. The scale bar either missing or not clearly visible for some pictures. Some captions do not include complete description. For instance, in fig 1-q1-1a6, the scale bar set is not reported. The use of dye for figs 1a and 1b not reported in the caption (it seems that it is yellow-dyed aqueous phase but it can easily get mixed with oil droplets).

R4.5. We increased the image size and fixed all the issues in the figures.

Q4.6. In Fig. 2 -7,8 droplets look polydisperse, but the surface area (A_n) reported in the plot C does not include any error bar for droplet values. Also, it seems the drops in fig2-1 to 2-6 form some structure, which is not explained in the text or caption.

R4.6. A_n is the surface area of the droplets as calculated from color images in 1-6. Yes, the SEM images are polydisperse, where the average drop size of $5.06 \pm 0.64 \mu m$ for DI water and $0.85 \pm 0.62 \mu m$ for 4.0 wt.% silica dispersion. We added the emulsion droplet size figure obtained from Cryo-SEM images in **Figure S16**. In color camera images we have the resolution of $1.86 \mu m/px$ and we cannot capture the single droplets in color images. A_n is the overall surface area of droplets as identified by dark spots in the images. We have clarified this point in the text as:

128 (Supporting information, section IV). The size of the emulsion droplet reduces upon adding
129 silica particles and increasing the concentration from $\sim 5 \mu m$ for DI water to $\sim 1 \mu m$ for 4.0
130 wt.% silica dispersion (**Fig. S16**), indicating the enhanced emulsification properties of the
131 system. The significantly lower interfacial tension in the presence of nanoparticles (**Figs. S1-2**
and **Figs. 2d-e**) attributes to the smaller emulsion droplet size (**Fig. S16**).
The time evolution of the dark zone at the surface of the aqueous droplet, normalized by the
droplet surface area, and denoted (A_n), at each time step is plotted in **Fig. 2c** (black for DI
water and red for a 4.0 wt.% silica dispersion). The polydispersity observed in emulsion droplet
size obtained from Cryo-SEM images (**Fig. S16**) is not detectable in the result obtained from
color camera images as in **Fig. 2c** we quantify the overall surface area of droplet. At the

Figure S16: Emulsion droplet size (droplet diameter) over silica concentration. The droplet size is obtained from Cryo-SEM microscopic images.

References:

Bala Subramaniam, Anand, Manouk Abkarian, L. Maha devan, and Howard A. Stone. "Non-spherical bubbles." *Nature* 438, no. 7070 (2005): 930-930.

Clegg P.S., "Bijels: Bicontinuous Particle-stabilized Emulsions", vol. 10. *Royal Society of Chemistry*, (2020): 1-33.

Culligan, K. A., Dorthe Wildenschild, Britt Stenhøj Baun Christensen, W. G. Gray, and M. L. Rivers. "Pore-scale characteristics of multiphase flow in porous media: A comparison of air–water and oil–water experiments." *Advances in Water Resources* 29, no. 2 (2006): 227-238.

Darby, Ronald, Ron Darby, and Raj P. Chhabra. Chemical engineering fluid mechanics, revised and expanded. *CRC Press*, 2017.

Gueven, Ibrahim, Stefan Frijters, Jens Harting, Stefan Luding, and Holger Steeb. "Hydraulic properties of porous sintered glass bead systems." *Granular Matter* 19, no. 2 (2017): 1-21.

Herzig, Eva M., K. A. White, Andrew B. Schofield, Wilson CK Poon, and Paul S. Clegg. "Bicontinuous emulsions stabilized solely by colloidal particles." *Nature Materials* 6, no. 12 (2007): 966-971.

Ioana Stanciu, Rheological Properties Mineral Oil (2011)

Liu, Tan, Yixuan Yin, Yang Yang, Thomas P. Russell, and Shaowei Shi. "Layer-by-layer engineered all-liquid microfluidic chips for enzyme immobilization." *Advanced Materials* 34, no. 5 (2022): 2105386.

López-Montilla, Juan Carlos, Paulo Emilio Herrera-Morales, Samir Pandey, and Dinesh O. Shah. "Spontaneous emulsification: Mechanisms, physicochemical aspects, modeling, and applications." *Journal of Dispersion Science and Technology* 23, no. 1-3 (2002): 219-268.

Stratford, Kevin, Ronjojoy Adhikari, Ignacio Pagonabarraga, J-C. Desplat, and Michael E. Cates. "Colloidal jamming at interfaces: A route to fluid-bicontinuous gels." *Science* 309, no. 5744 (2005): 2198-2201.

REVIEWERS' COMMENTS

Reviewer #1 (Remarks to the Author):

The authors have taken some but not all of the advice given in the first review of this manuscript. In particular, they more clearly differentiate their work from prior art. This is a welcome improvement. However, they have not demonstrated the generality across different compositions of emulsions, etc. Instead, they focused on providing more technical information on the initial design. I suspect the potential for impact as a result will be lower than it could have been. I do not think that should preclude the authors from moving forward in publishing the work in Nature Communications. Before that happens, I would only recommend two things be addressed:

1) Some of the figures lack finesse and are not up to the artistic standards of this journal. Please revise accordingly. Specifically, Figure 3. The fonts used throughout the figures also should stick to those recommended by the journal, or at least consistent with those used in the text of the publication.

2) The authors should make mention of 3D printing of hierarchically porous materials from liquid photopolymer resins undergoing microphase separation in-situ. Pavel Levkin's work in particular. These are essentially complementary to those considered in this work, which are soft, whereas those from the Levkin group are hard. Ref here:

Dong, Z., Cui, H., Zhang, H. et al. 3D printing of inherently nanoporous polymers via polymerization-induced phase separation. Nat Commun 12, 247 (2021).
<https://doi.org/10.1038/s41467-020-20498-1>

Reviewer #3 (Remarks to the Author):

The authors have made substantial revisions, which has significantly improved manuscript.

The major concern is labeling the morphology of Figure R1 bicontinuous. I agree that a bicontinuous morphology is one in which two separate phases are independently connected to form a single phase, but the images shown in Figure R1 have isolated oil domains. Therefore, the morphology is more similar to an oil-in-water emulsion. What seems to be happening is that there is phase coexistence between an emulsion and a macrophase separated mixture. The authors should make sure that their morphology labeling is correct.

We appreciate the feedback provided by the reviewers, which has certainly helped us to improve the revised manuscript. We addressed each points in the revised version as follow:

Reviewer #1 (Remarks to the Author):

The authors have taken some but not all of the advice given in the first review of this manuscript. In particular, they more clearly differentiate their work from prior art. This is a welcome improvement. However, they have not demonstrated the generality across different compositions of emulsions, etc. Instead, they focused on providing more technical information on the initial design. I suspect the potential for impact as a result will be lower than it could have been. I do not think that should preclude the authors from moving forward in publishing the work in Nature Communications. Before that happens, I would only recommend two things be addressed:

1) Some of the figures lack finesse and are not up to the artistic standards of this journal. Please revise accordingly. Specifically, Figure 3. The fonts used throughout the figures also should stick to those recommended by the journal, or at least consistent with those used in the text of the publication.

We made the fonts of all figures uniform (according to figure check list provided by the Nature Communications journal) and increased the quality of the figures.

2) The authors should make mention of 3D printing of hierarchically porous materials from liquid photopolymer resins undergoing microphase separation in-situ. Pavel Levkin's work in particular. These are essentially complementary to those considered in this work, which are soft, whereas those from the Levkin group are hard. Ref here:

Dong, Z., Cui, H., Zhang, H. et al. 3D printing of inherently nanoporous polymers via polymerization-induced phase separation. Nat Commun 12, 247 (2021). <https://doi.org/10.1038/s41467-020-20498-1>

Thank you for the suggestion, we discussed the paper in the introduction as:

3D printing techniques, in combination with polymerization-induced phase separation, en-
able the formation of microstructures within the 3D printed solid structures in air [27]. In these
methods, after printing a multicomponent mixture in the desired configuration, the polymer-
ization, which increases the molecular weight of one of the components, starts and initiates
the phase segregation. Internal textures are created within the printed frame during polymer-
ization as they get solidified [27,28]. Structured liquids with submicrometer domains can be

Reviewer #3 (Remarks to the Author):

The authors have made substantial revisions, which has significantly improved manuscript.

The major concern is labeling the morphology of Figure R1 bicontinuous. I agree that a bicontinuous morphology is one in which two separate phases are independently connected to form a single phase, but the images shown in Figure R1 have isolated oil domains. Therefore, the morphology is more similar to an oil-in-water emulsion. What seems to be happening is that there is phase coexistence between an emulsion and a macrophase separated mixture. The authors should make sure that their morphology labeling is correct.

We agree with the reviewer that in some sections we have separated oil domains which may deviate from the definition of the bicontinuous phase. To avoid further confusion, we removed this terminology through the manuscript and used the interconnected zones of oil and emulsion phases.